# GRAPH CONVOLUTIONAL NETWORKS VIA ADAPTIVE FILTER BANKS

## ABSTRACT

Graph convolutional networks have been a powerful tool in representation learning of networked data. However, most architectures of message passing graph convolutional networks (MPGCNs) are limited as they employ a single message passing strategy and typically focus on low-frequency information, especially when graph features or signals are heterogeneous in different dimensions. Then, existing spectral graph convolutional operators lack a proper sharing scheme between filters, which may result in overfitting problems with numerous parameters. In this paper, we present a novel graph convolution operator, termed BankGCN, which extends the capabilities of most MPGCNs beyond single 'low-pass' features and simplifies spectral methods with a carefully designed sharing scheme between filters. BankGCN decomposes multi-channel signals on arbitrary graphs into subspaces and shares adaptive filters to represent information in each subspace. The filters of all subspaces differ in frequency response and together form a filter bank. The filter bank and the signal decomposition permit to adaptively capture diverse spectral characteristics of graph data for target applications with a compact architecture. We finally show through extensive experiments that BankGCN achieves excellent performance on a collection of benchmark graph datasets.

## 1 INTRODUCTION

In many application domains, structured data is supported by graphs or networks. To deal with such data, a collection of graph convolutional networks (GCNs), including MPGCNs and spectral GCNs, have been proposed by generalizing architectures used for data in the Euclidean domain, to the irregular graphs. Unfortunately, both types of architectures have implicit limitations when it comes to efficiently representing graph data information in terms of frequency filtering.

MPGCNs (Gilmer et al., 2017; Velikovi et al., 2018; Hamilton et al., 2017; Xu et al., 2019) built on diverse message passing (MP) schemes are much prevalent with their flexible and intuitive formulations of graph convolution. The message passing schemes in classical MPGCNs, however, mostly focus on the low frequency characteristics of the data. For example, GCN (Kipf & Welling, 2017) performs as Laplacian smoothing (Li et al., 2018), and the mean aggregator adopted in GraphSage (Hamilton et al., 2017) has naturally low-pass properties. However, in addition to low-frequency components, graph data may also carry rich information in the middle- and high-frequency ranges. There have been some recent attempts to complement low-pass features, either with predefined filters (Gao et al., 2019; Chang et al., 2021) or with other aggregation schemes (Abu-El-Haija et al., 2019; Zhu et al., 2020). Yet, they still have some limitations, such as the nonadaptiviy of predefined filters. The importance of different frequency components may vary with the target tasks, it is thus beneficial to be able to adapt the representation accordingly.

Spectral graph convolutional networks (Defferrard et al., 2016; Levie et al., 2018; Bianchi et al., 2021) are designed in the graph frequency domain directly and are much more powerful alternatives to process diverse spectral characteristics of graphs. The lack of proper sharing schemes between filters, however, renders these models redundant and even prone to overfitting.

We go beyond the above limitations and propose to build effective adaptive representations for graph data with diverse spectral properties. We design a novel graph convolution operator termed BankGCN that utilizes an adaptive filter bank to process graph data in the frequency domain, as presented in Fig. 1. BankGCN provides an effective implementation to simplify spectral methods

with a proper sharing scheme between filters with the help of signal decomposition. Firstly, we decompose multi-channel graph signals into a collection of subspaces through projection, in order to adaptively separate input data according to signal characteristics. Components in a subspace will then share a learnable filter that captures their particular frequency properties. Notably, representations are built on finite impulse response (FIR) filters with a universal design (Tremblay et al., 2018) that correspond to local message passing schemes in the spatial domain. Filters of all the subspaces together form a filter bank, and they are simultaneously learned from data together with subspace decomposition. The filter bank representation is regularized to favour diversity in the frequency responses, in order to properly capture various spectral components in the graph signals. The proposed convolution operator is stackable, and can be optimized together with other modules in the GCNs like graph pooling. We validate the proposed architecture through extensive experiments on various graph classification tasks. BankGCN is more powerful than most MPGCNs, such as GCN (Kipf & Welling, 2017), GraphSage (Hamilton et al., 2017), and GIN (Xu et al., 2019), in that it is able to exploit more diverse spectral characteristics than 'low-pass' features in the data and adapts to its heterogeneous properties with *a group of learnable multi-hop* message passing strategies. Furthermore, it outperforms its counterpart ChebNets (Defferrard et al., 2016) with a more compact architecture and achieves better generalization, and is superior to the most recent spectral method ARMA (Bianchi et al., 2021). It would be interesting to extend BankGCN to tasks like link prediction and applications into non-Euclidean data like 3-D point clouds.

## 2    RELATED WORK

We briefly overview the main architectures for graph representation learning. We describe first existing message passing schemes, and then discuss several spectral methods.

**Message Passing Graph Convolutional Networks.** MPGCNs design graph convolution with a variety of message passing schemes in the spatial domain. For instance, messages are aggregated with the node-wise mean or max operation in a localized neighborhood in GraphSage (Hamilton et al., 2017), or based on attention scores in GAT (Velikovi et al., 2018). A more expressive scheme, GIN, is further proposed with summing features in a neighborhood followed by a multi-layer perceptrons (MLP) to approximate any injective function on multiset (Xu et al., 2019). However, these methods are typically constrained to a single one-hop message passing strategy that mostly captures 'low-pass' characteristics in the graph data. Klicpera et al. (2019) expand the MP range to multi-hop neighborhoods with graph diffusion operators. Besides, several methods are proposed to implement an equivalent high-order (polynomial) filter with the whole network to alleviate over-smoothing representation, including SGC (Wu et al., 2019), LGC (Navarin et al., 2020), GPR-GNN (Chien et al., 2021), SGF (NT et al., 2021), and S$^2$GC (Zhu & Koniusz, 2021). However, a single equivalent filter may be limited when representing multi-channel signals with diverse frequency characteristics. Alternatively, a group of works attempt to complement the 'low-pass' features using different strategies, such as graph wavelet filters (Gao et al., 2019; Chang et al., 2021), a one-order high-pass filter (Bo et al., 2021) and an additional quadratic middle-pass filter (Wu et al., 2021), and other aggregation schemes (Abu-El-Haija et al., 2019; Zhu et al., 2020). Contrary to these methods, we employ a learnable filter bank with various frequency responses in a single layer to adaptively capture diverse spectral characteristics of signals rather than merely adding 'high-pass' features or specific components via predefined graph wavelets. Finally, in contrast with some works that further consider adaptive filters (Li et al., 2021; Dong et al., 2021; Pasa et al., 2021), BankGCN is a more powerful and compact design with polynomial filter banks to avoid eigen-decomposition and a further sharing scheme between filters to reduce free parameters. More detailed comparisons in App. F.

**Filtering on Graphs and Spectral GCNs.** Filtering and Graph Fourier Transform have been generalized to graph data via spectral graph theory (Shuman et al., 2013). Correspondingly, several spectral GCNs are derived from filtering in the graph frequency domain directly, as pioneered by Bruna et al. (2014). However, they rely on computationally expensive eigen-decomposition of the graph Laplacian. To avoid the eigendecomposition, similarly to dictionary learning methods (Thanou et al., 2014) in graph signal processing (GSP), the filters are built as polynomial or rational functions of eigenvalues of the graph Laplacian, such as Chebyshev polynomials in ChebNets (Defferrard et al., 2016) and TIGraNet (Khasanova & Frossard, 2017), Cayley polynomials in CayleyNets (Levie et al., 2018), and auto-regressive moving average (ARMA) filters (Bianchi et al., 2021). In contrast with these spectral convolution methods that focus on the implementation of filters with desirable prop-

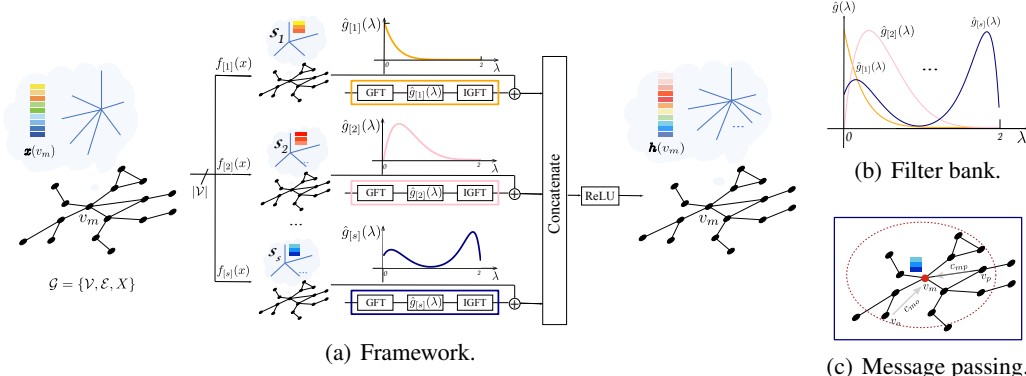

(a) Framework.

(b) Filter bank.

(c) Message passing.

Figure 1: Illustration of the BankGCN operator. In each layer, the graph signal $\boldsymbol{x}(v_m)$ is decomposed into a collection of subspaces and processed by an adaptive filter bank to capture distinct frequency properties. A filter in the filter bank corresponds to an adapted $K$-hop neighborhood message passing scheme in the spatial domain, as illustrated in (b) and (c).

erties, like localization and narrowband specialization, this paper focuses on the design of the filter bank in order to construct a simplified architecture with improved generalization.

Finally, there are several papers in graph signal processing literature about the design of graph filter banks for graph signal decomposition (Narang & Ortega, 2012; Tanaka & Sakiyama, 2014) and multiscale analysis (Hammond et al., 2011; Narang & Ortega, 2013). However, they usually work under a rigid constraint, e.g., perfect reconstruction, and thereby their produced (sparse) representations may not be flexible enough to adapt to diverse tasks in machine learning. In contrast, our method relaxes the perfect reconstruction condition and rather regularizes filters in a filter bank to be sufficiently different in the graph frequency domain.

## 3 PRELIMINARIES AND PROBLEMS

In this paper, we consider data that is represented on undirected graphs. Each vertex of graphs is attributed a $d$-channel signal or feature. The node signal is typically multi-channel, *i.e.*, $d > 1$, and with different spectral statistics along different dimensions, which we refer to heterogeneous signals in this paper. Notations and preliminaries on Graph Fourier Transform and filtering are introduced in App. A.

We first show how filtering can be implemented by messaging passing, which is used in many state-of-the-art graph representation learning methods. Filtering in the graph frequency domain and MP in the spatial domain is closely related (Shuman et al., 2013). Taking the most popular GCN (Kipf & Welling, 2017) as an example, according to its initial formation, the $j$-th channel of a filtered signal is

$$\boldsymbol{h_j} = \sum_{i=1}^{d}(I + D^{-\frac{1}{2}}AD^{-\frac{1}{2}})\Theta\boldsymbol{x_i} = \sum_{i=1}^{d}(2I - L)\Theta_{i,j}\boldsymbol{x_i} = \sum_{i=1}^{d}\Theta_{i,j}U\hat{g}(\Lambda)U^*\boldsymbol{x_i} \qquad (1)$$

where $\Theta$ denotes learnable parameters, $d$ indicates the dimension of signals, and $\hat{g}(\Lambda) = 2I - \Lambda$ is a low-pass filter (NT & Maehara, 2019; Min et al., 2020). The renormalized version of GCN and the GIN can be formulated similarly. Please refer to (Balcilar et al., 2021) for their specific equivalent supports of $\hat{g}(\Lambda)$. Besides, GraphSage uses the mean or max operator to aggregate information in neighborhoods. Although it is hard to explicitly formulate its corresponding frequency response, the element-wise mean operation is a low-pass operation by nature and the element-wise max operator performs as an envelope extraction that suppresses the high-frequency components.

Another line of works, spectral GCNs, are directly derived from spectral filtering. For the representative ChebNets and CayleyNets, the filters are defined as $\hat{g}(\lambda) = \sum_{k=0}^{K}\theta^{(k)}T_k(\lambda)$ in the graph frequency domain with the Chebyshev polynomial or Cayley polynomial basis $\{T_k(\cdot)\}$ and correspondingly

$$\boldsymbol{h_j} = \sum_{i=1}^{d}U\sum_{k=0}^{K}\Theta_{i,j}^{(k)}T_k(\Lambda)U^*\boldsymbol{x_i} = \sum_{i=1}^{d}U\hat{g}_{ij}(\Lambda)U^*\boldsymbol{x_i}, \qquad (2)$$

where $K$ is the order of the polynomial filters and $\{\Theta^{(k)}\}$ are learnable parameters.

Notably, the above MPGCNs employ just a single kind of filter to handle all the channels of signals, and then takes different (linear) combinations of the filtered signals to obtain different channels of the output signals. Thereby, most MPGCNs are restricted in the number and types of filters and have a limited capacity in frequency-domain filtering. On the contrary, ChebNets and CayleyNets employ a different filter for each mapping from an input channel to an output channel and adopt a total of $d \times d'$ $K$-order polynomial filters ($d$ and $d'$ denote the respective number of channels of input and output signals) , as presented in Eq. 2. This naive design neglects the potential relationship between filters and fails to reduce redundancy. Consequently, they explore such a large number of learnable filters to process diverse frequency components of graph data at the expense of numerous free parameters, which sometimes leads to overfitting.

## 4 THE BANKGCN ALGORITHM

In this section, we first outline the framework of BankGCN, then present its main elements, and finally discuss connections with spectral methods.

### 4.1 GRAPH CONVOLUTION WITH ADAPTIVE FILTER BANKS

Signals (features) supported on nodes of a graph are usually high-dimensional and composed of multiple spectral patterns. In other words, different signal channels vary differently over nodes, leading to diverse spectral characteristics. To handle the heterogeneous signals, we employ a filter bank composed of a set of filters with different frequency responses in the design of BankGCN. In order to further facilitate information processing, signal decomposition is adopted to explore latent relationships in the data.

For a multi-channel input node signal $\boldsymbol{x}(v_m)$, we first adaptively decompose it into different subspaces and then employ different filters to deal with the signal components in each subspace separately. The decomposition aims to map the components of signals with similar characteristics into the same subspace in order to facilitate the subsequent adaptive filtering. This is implemented with learnable subspace projections. Mathematically, the input signal $\boldsymbol{x}(v_m)$ is projected into $s$ subspaces with a group of projection functions denoted by $\{f_{[p]}(\cdot)\}$,

$$\boldsymbol{r}_{[p]}(v_m) = f_{[p]}(\boldsymbol{x}(v_m)), \quad p = 1, 2, \ldots, s, \ \forall v_m \in \mathcal{V}. \tag{3}$$

Here, the subscript "$[p]$" indicates the terms belonging to the $p$-th subspace and $\boldsymbol{r}_{[p]}(v_m)$ is the projected signal. The choices for projection functions will be introduced in Section 4.2. Subsequently, an adaptive filter $g_{[p]}(\cdot)$ is designed within each subspace to represent the spectral characteristics of the corresponding signal components,

$$\boldsymbol{h}_{[p]}(v_m) = (g_{[p]} * \boldsymbol{r}_{[p]})(v_m), \tag{4}$$

where we reuse '$*$' to denote the convolution between each channel of signal $\boldsymbol{r}_{[p]}$ and the filter $g_{[p]}$.

The filtered signals $\boldsymbol{h}_{[p]}(v_m)$, $p = 1, 2, \cdots, s$, of all the subspaces are concatenated to produce the output features. A Rectified Linear Unit (ReLU) is then used as a non-linear activation function,

$$\boldsymbol{h}(v_{l,m}) = \text{Concat}(\boldsymbol{h}_{[1]}(v_{l,m}), \boldsymbol{h}_{[2]}(v_{l,m}), \ldots, \boldsymbol{h}_{[s]}(v_{l,m})), \tag{5}$$

$$\boldsymbol{x}(v_{l+1,m}) = \text{ReLU}(\boldsymbol{h}(v_{l,m})), \tag{6}$$

where the subscript $l$ indicates variables or parameters in the $l$-th layer to describe the forward propagation between different layers in a hierarchical architecture.

Specially, considering that some filters are only supported on medium-to-high frequency components that usually mean large signal variations in the spatial domain, we further introduce a shortcut in each subspace that corresponds to full-pass in the graph frequency domain in order to make the filtered signals stable. Correspondingly, the filtered signal $\boldsymbol{h}_{[p]}(v_m)$ in each subspace becomes

$$\boldsymbol{h}_{[p]}(v_m) = (g_{[p]} * \boldsymbol{r}_{[p]})(v_m) + \boldsymbol{r}_{[p]}(v_m). \tag{7}$$

The shortcut can further facilitate back propagation of gradients, as for CNNs (He et al., 2016).

The proposed graph convolution operator is stackable, and the subspace mapping function in the following layers will further combine features from different subspaces in the preceding layers to enable information interaction between different channels.

## 4.2 SUBSPACE PROJECTION

For the projection function $f_{[p]}(\cdot)$, there is a variety of design choices. Here, we take the linear mapping as an example, since it is simple yet able to decompose different channels of signals. Specifically, the $d$-channel [1] signal $\boldsymbol{x}(v_m)$ is projected into $s$ different subspaces with learnable matrices $\{W_{[p]}\}_{p=1}^s$. For the sake of simplicity, all the subspaces have the same dimension in this paper. To increase flexibility, we further introduce a learnable bias $\boldsymbol{b}_{[p]}$ for each subspace.

$$\boldsymbol{r}_{[p]}(v_m) = f_{[p]}(x(v_m)) = W_{[p]}^T \boldsymbol{x}(v_m) + \boldsymbol{b}_{[p]}, \quad p = 1, 2, \ldots, s. \tag{8}$$

The introduction of subspace projection brings in two advantages: (i) it simplifies the learning process. Since the multi-channel graph signals have been decomposed, the filter in each subspace just needs to learn to capture spectral characteristics of the corresponding signal components; (ii) with low-dimensional subspaces, it limits the dimension of features output by the filter bank, and thereby reduces the number of free parameters and computation in the following layers.

## 4.3 FILTERS

In order to handle graphs with arbitrary topologies and diverse signals, we adopt universal and adaptive filters to construct the filter bank.

For any function $t : \mathcal{D} = [0, 2] \to \mathbb{R}$, we can obtain a corresponding filter whose frequency response is $\hat{g}_{[p]}(\lambda) = t(\cdot)$, and its spatial construction is computed through IGFT (defined as Eq. S2 in App. A):

$$g_{[p]}(v_m) = \sum_{i=1}^n \hat{g}_{[p]}(\lambda_i) u_i(v_m), \tag{9}$$

with $\boldsymbol{u_i}$, $i = 1, \cdots, n$, the eigenvectors of the symmetric normalized graph Laplacian of a graph. Notably, we directly design the frequency response of the filter for the continuous range $\mathcal{D} = [0, 2]$ in which the spectrum of an arbitrary graph locates as introduced in Section A. In other words, given the discrete spectrum $\{\lambda_0, \lambda_1, \ldots, \lambda_n\}$ of an arbitrary graph, we have $\lambda_i \in \mathcal{D}$ and obtain the corresponding filter value on these specific discrete values $\hat{g}_{[p]}(\lambda_i) = t(\lambda_i)$, for $i = 1, 2, \ldots, n$. Thereby, the filter $g_{[p]}(\cdot)$ is adaptable to any graph even with different topologies, *i.e.*, it is a universal form (Tremblay et al., 2018; Levie et al., 2019).

Furthermore, we constrain the filter to the $K$-order polynomial function space in order to avoid the computation-intense eigendecomposition of the graph Laplacian, similarly to parametric dictionary learning (Thanou et al., 2014) in GSP. It corresponds to an FIR filter. Mathematically, the frequency response of a filter can be represented as

$$\hat{g}_{[p]}(\lambda) = \sum_{k=0}^K \alpha_{[p]}^{(k)} T_k(\lambda), \tag{10}$$

where $\{T_k\}$ denotes a specific polynomial basis such as Chebyshev polynomials, and $\{\alpha_{[p]}^{(k)}\}$ indicates the corresponding coefficients. With $\{\alpha_{[p]}^{(k)}\}$ learnable, we obtain an adaptive filter whose frequency response adapts to the data and to the target task. Correspondingly, for the signal projected to the $p$-th subspace $\boldsymbol{r}_{[p]}(v_m)$ and $R_{[p]} = [\boldsymbol{r}_{[p]}(v_1), \boldsymbol{r}_{[p]}(v_2), \ldots, \boldsymbol{r}_{[p]}(v_n)]^T$, the filtered signal is calculated as

$$\boldsymbol{h}_{[p]}(v_m) = (\boldsymbol{r}_{[p]} * g_{[p]})(v_m) = \sum_{k=0}^K \alpha_{[p]}^{(k)} (T_k(L) R_{[p]})_m^T. \tag{11}$$

Equivalently, according to the relationship between frequency filtering and localized linear transforms in the spatial domain (Shuman et al., 2013), the filtering strategy corresponds to a message passing scheme within a $K$-hop neighborhood

$$\boldsymbol{h}_{[p]}(v_m) = c_{mm} \boldsymbol{r}_{[p]}(v_m) + \sum_{v_o \in N^K(v_m)} c_{mo} \boldsymbol{r}_{[p]}(v_o), \tag{12}$$

---

[1] For the input signal with $d = 1$, we set $s$ as the feature channels in the neural network in the first layer so that the signal is scaled differently in each subspace. In the following hidden layers, it can be used as the other cases to handle multi-channel features produced by previous layers in GCNs.

with

$$c_{mo} = \sum_{k=0}^{K} \alpha_{[p]}^{(k)} (T_k(L))_{m,o} \quad \forall m, o \in \{1, 2, \dots, n\}. \tag{13}$$

Specially, the induced $K$-hop message passing scheme is learned from the data and exploits the multi-hop topological information of graphs through polynomials of the graph Laplacian. Signal information is also taken into consideration through the learnable parameters $\{\alpha_{[p]}^{(k)}\}$ as well as in the subspace projection step. More importantly, it permits to represent features that do not only have "low-pass" properties and explores the frequency components in a data-driven manner.

### 4.4 DIVERSITY REGULARIZATION FOR FILTER BANKS

The filters constituting a filter bank should ideally have diverse frequency responses so that the signal is decomposed through filtering into a series of signals with different frequency characteristics. In GSP, the filters are usually band-pass and divide the spectrum in different bands. Considering that strict band-pass filters are difficult to fit through polynomial functions, we relax this strict band-pass requirement and rather target filters with diverse frequency responses, which we call "diversity condition". With the filter $\hat{g}_{[p]}(\lambda)$ given as a $K$-order polynomial function, the regularization on the filter is imposed on the respective polynomial coefficients $\{\alpha_{[p]}^{(k)}\}_{p,k}$ in Eq. 10. To achieve the diversity condition, we regularize the polynomial coefficients to be well distributed in the parameter space. Considering that the distances of the coefficient vectors of two scaled filters may still be large in terms of the Euclidean distance, we thereby take the cosine distance to measure the distance between the polynomial coefficients of filters. Specifically, the regularization term is:

$$\Omega(\alpha) = \max_{p \neq q} \frac{|<\boldsymbol{\alpha}_{[p]}, \boldsymbol{\alpha}_{[q]}>|}{\|\boldsymbol{\alpha}_{[p]}\|_2 \|\boldsymbol{\alpha}_{[q]}\|_2}, \tag{14}$$

where $\boldsymbol{\alpha}_{[p]} = [\alpha_{[p]}^{(0)}, \alpha_{[p]}^{(1)}, \dots, \alpha_{[p]}^{(K)}]^T$. The max function reflects the maximum similarity between the polynomial coefficients of any pair of filters in the filter bank. Through minimizing Eq. 14, the most similar filters will have different orientations in the parameter space. Thus, all the pairs of filters tend to be different. When $\{T_k(L)\}_k$ is an orthogonal basis such as Chebyshev polynomials, the diversity of polynomial coefficients $\{\boldsymbol{\alpha}_{[p]}\}_p$ implies that filters defined as Eq. 10 are different in the graph frequency domain. More intuitively, the message passing schemes in the spatial domain induced by the filters are different with diverse $\{\boldsymbol{\alpha}_{[p]}\}_p$, as presented in Eq. 12 and Eq. 13.

We can note that, if the filter is defined on the basis composed of rectangular pulse functions[2], *i.e.*,

$$T_k(\lambda) = \begin{cases} 1 & \frac{2k}{K+1} \leq \lambda < \frac{2(k+1)}{K+1} \\ 0 & \text{others} \end{cases}, \tag{15}$$

the ideal subband filter banks, whose filters have different passbands in GSP, just corresponds to the optimal solution to the regularization with $\Omega(\alpha) = 0$ in Eq. 14, when $K \geq s$.

With $\mathscr{T}_\Theta(\mathcal{G}, Y)$ generally representing a target function, the overall objective function is then formulated as

$$\min_\Theta \; \mathscr{T}_\Theta(\mathcal{G}, Y) + \gamma \, \Omega(\alpha), \tag{16}$$

where $Y$ indicates ground truth labels, $\Theta$ denotes the parameter set including $\{\boldsymbol{\alpha}_{[p]}, W_{[p]}, \boldsymbol{b}_{[p]}\}_{p=1}^s$, and $\gamma$ is a hyperparameter to adjust the contribution of regularization term. Like most popular graph convolution operators, BankGCN can be optimized via gradient-based methods together with modules such as graph pooling operators in GCNs. It achieves linear computational complexity with $O(K|\mathcal{E}|d)$ and constant learning complexity, similarly to most existing MPGCNs.

### 4.5 DISCUSSION

BankGCN actually provides an effective simplification of existing polynomial-based spectral methods with parameter decomposition. With $\Theta = [\Theta_0; \Theta_1; \dots; \Theta_K] \in \mathbb{R}^{d \times d' \times (K+1)}$ representing the

---

[2]Note that this is not the case in this paper.

free parameters of the spectral methods adopting corresponding polynomial filters, BankGCN with a linear mapping as projection function actually provides a decomposition based approximation of $\Theta$ with

$$[W_{[1]} \otimes \boldsymbol{\alpha}_{[1]}, W_{[2]} \otimes \boldsymbol{\alpha}_{[2]}, \ldots, W_{[s]} \otimes \boldsymbol{\alpha}_{[s]}], \quad (17)$$

where $\otimes$ indicates outer product. This decomposition strategy allows for filter sharing to reduce potential redundancy of existing spectral methods. Multi-channel signals are first adaptively decomposed into various subspaces with $\{W_{[p]}\}_p$, and signal components within a subspace share a filter defined with $\boldsymbol{\alpha}_{[p]}$ to process corresponding characteristics. In addition, it permits to further control the relationship between filters with the diversity regularization as introduced in Section 4.4.

We further study the capacity of BankGCN in handling the spectrum of signals with a compact architecture.

**Proposition 1.** *BankGCN can capture and preserve diverse frequency components of input graph signals, like its counterpart introduced in Eq. 2.*

*Proof.* Please refer to App. B. □

As demonstrated in Prop. 1 and its proof, a single filter $g_{[p]}$ in BankGCN together with signal projection is capable to explore and represent diverse frequency components of graph signals, as done in its counterpart with a group of filters $\{g_{ij}\}_i$. This theoretically validates the effectiveness of BankGCN in reducing the number of filters and thereby potential redundancy in existing polynomial-based spectral architectures, while maintaining the capability of representing diverse spectral information.

## 5 EXPERIMENTS

In this section, in order to evaluate models in learning representation of diverse spectral information of graph data, we resort to node classification and graph classification tasks.

**Baselines.** We compare BankGCN with several state-of-the-art graph convolution methods, including MPGCNs and spectral methods, as listed in Table 1 and Table 2. Furthermore, two simplifications of the proposed model (BankGCN-Diff and BankGCN-NR) are considered for ablation study. Please refer to App. C & D for details.

### 5.1 NODE CLASSIFICATION

We first evaluate the proposed model in node classification tasks on several heterophilic (*i.e.*, neighbor nodes are dissimilar in labels/features) graph datasets, in order to validate the advantages of diverse frequency representation with filter banks. We follow the data split and evaluation procedure of (Pei et al., 2020; Zhu et al., 2020; Li et al., 2021), and describe the detail information of datasets and experimental settings in App. C. Results of baselines are cited from their respective publications if available, or recent public works (Zhu et al., 2020; Li et al., 2021). The metric $\beta$ introduced by Pei et al. (2020) reveals the degree of homophily of graph data in datasets (Small $\beta$ cor-

Table 1: Results (mean accuracy $\pm$ stdev) on node classification with different data splits (* and † denoting the results cited from (Zhu et al., 2020) and (Li et al., 2021), respectively).

| $\beta$ | Texas 0.06 | Wisconsin 0.16 | Cornell 0.11 | Actor 0.24 |
|---|---|---|---|---|
| GCN* | $59.46 \pm 5.25$ | $59.80 \pm 6.99$ | $57.03 \pm 4.67$ | $30.26 \pm 0.79$ |
| SGC † | $58.9 \pm 6.1$ | $51.8 \pm 5.9$ | $58.1 \pm 4.6$ | - |
| GAT* | $53.38 \pm 4.45$ | $55.29 \pm 8.71$ | $58.92 \pm 3.32$ | $26.28 \pm 1.73$ |
| GEOM-GCN | $67.57$ | $64.12$ | $60.81$ | $31.63$ |
| MixHop* | $77.84 \pm 7.73$ | $75.88 \pm 4.90$ | $73.51 \pm 6.34$ | $32.22 \pm 2.34$ |
| GraphSage* | $82.43 \pm 6.14$ | $81.18 \pm 5.56$ | $75.95 \pm 5.01$ | $34.23 \pm 0.99$ |
| ChebNets† | $79.7 \pm 5.0$ | $82.5 \pm 2.8$ | $76.5 \pm 9.4$ | - |
| ARMA † | $82.2 \pm 5.1$ | $78.4 \pm 4.6$ | $74.9 \pm 2.9$ | - |
| SGF | $80.56 \pm 0.63$ | $87.06 \pm 4.66$ | $82.45 \pm 6.19$ | - |
| H2GCN* | $84.86 \pm 6.77$ | $86.67 \pm 4.69$ | $82.16 \pm 4.80$ | $35.86 \pm 1.03$ |
| ASGAT-Cheb | $85.1 \pm 5.7$ | $86.3 \pm 3.7$ | $82.7 \pm 8.3$ | - |
| BankGCN | $\mathbf{86.49 \pm 4.19}$ | $\mathbf{87.06 \pm 3.19}$ | $\mathbf{83.78 \pm 4.68}$ | $\mathbf{36.87 \pm 0.95}$ |

responding to strong heterophily). As presented in Table 1, BankGCN achieves the best performance on all the datasets. In contrast, the MPGCNs that rely on low-pass features, like GCN and GAT, perform worse. Notably, BankGCN still outperforms several of the most recent MPGCNs that improve on the over-smoothing issues, including a simplified GCN (SGC), improved aggregation schemes MixHop, GEOM-GCN and H2GCN, and adaptive filter based methods SGF and ASGAT.

Table 2: Results on graph classification with 20 runs for different datasets (#P/L denotes the number of free parameters per hidden layer).

| | ENZY | DD | NCI1 | PROT | NCI109 | MUTA | FRAN | $\sim$#P/L |
|---|---|---|---|---|---|---|---|---|
| GCN | $62.75 \pm 5.83$ | $77.75 \pm 3.55$ | $79.00 \pm 1.93$ | $74.87 \pm 4.08$ | $78.90 \pm 1.52$ | $81.34 \pm 1.61$ | $62.21 \pm 2.41$ | 4.2k |
| GraphSage | $66.75 \pm 6.31$ | $75.21 \pm 2.72$ | $80.97 \pm 1.87$ | $75.13 \pm 4.04$ | $79.54 \pm 2.24$ | $82.30 \pm 1.48$ | $63.91 \pm 1.96$ | 8.3k |
| GIN | $61.08 \pm 4.92$ | $75.42 \pm 3.31$ | $81.19 \pm 2.27$ | $74.91 \pm 3.88$ | $80.71 \pm 2.38$ | $81.66 \pm 2.48$ | $68.11 \pm 2.09$ | 8.3k |
| GAT | $62.67 \pm 7.52$ | $77.50 \pm 2.14$ | $79.43 \pm 2.38$ | $75.09 \pm 4.05$ | $79.16 \pm 1.85$ | $81.28 \pm 2.20$ | $63.89 \pm 1.53$ | 4.3k |
| ChebNets ($K = 2$) | $66.75 \pm 4.79$ | $77.67 \pm 2.91$ | $81.80 \pm 2.35$ | $74.64 \pm 4.75$ | $81.27 \pm 1.89$ | $82.50 \pm 1.58$ | $68.35 \pm 2.65$ | 12.4k |
| ARMA($K = 2$) | $63.33 \pm 6.32$ | $\mathbf{78.81 \pm 3.04}$ | $80.86 \pm 2.58$ | $74.91 \pm 5.36$ | $80.12 \pm 1.89$ | $81.97 \pm 1.81$ | $67.65 \pm 2.35$ | 12.4k |
| BankGCN-Diff ($s = 8$) | $66.92 \pm 5.71$ | $77.88 \pm 2.81$ | $80.07 \pm 2.03$ | $74.87 \pm 4.21$ | $79.23 \pm 2.29$ | $82.27 \pm 2.00$ | $64.63 \pm 1.96$ | 4.2k |
| BankGCN-NR ($K = 2, s = 8$) | $65.83 \pm 6.66$ | $77.03 \pm 4.08$ | $81.89 \pm 1.95$ | $75.36 \pm 4.68$ | $81.03 \pm 1.95$ | $82.44 \pm 1.69$ | $67.82 \pm 2.30$ | 4.2k |
| BankGCN ($K = 2, s = 8$) | $\mathbf{68.00 \pm 5.23}$ | $78.14 \pm 2.81$ | $\mathbf{82.06 \pm 1.75}$ | $75.67 \pm 4.19$ | $81.54 \pm 2.13$ | $\mathbf{82.89 \pm 1.61}$ | $67.82 \pm 2.30$ | 4.2k |
| BankGCN ($K = 2, s = 16$) | $66.83 \pm 5.19$ | $77.42 \pm 3.50$ | $81.93 \pm 2.15$ | $\mathbf{76.12 \pm 5.08}$ | $81.62 \pm 1.87$ | $82.57 \pm 1.61$ | $\mathbf{68.43 \pm 1.98}$ | 4.2k |

Table 3: Study on the order $K$ of filters and the number of subspaces $s$ per layer.

| | | ENZY | DD | NCI1 | PROT | NCI109 | MUTA | FRAN |
|---|---|---|---|---|---|---|---|---|
| | $s = 1$ | $63.58 \pm 6.31$ | $76.40 \pm 2.34$ | $80.46 \pm 2.34$ | $74.38 \pm 4.80$ | $79.23 \pm 2.29$ | $82.09 \pm 1.51$ | $65.52 \pm 2.44$ |
| $K = 2$ | $s = 4$ | $66.75 \pm 5.61$ | $78.14 \pm 2.81$ | $81.62 \pm 1.84$ | $75.67 \pm 4.61$ | $81.19 \pm 2.08$ | $82.70 \pm 1.63$ | $67.48 \pm 2.09$ |
| | $s = 8$ | $\mathbf{68.00 \pm 5.23}$ | $78.14 \pm 2.81$ | $\mathbf{82.06 \pm 1.75}$ | $75.67 \pm 4.19$ | $81.54 \pm 2.13$ | $\mathbf{82.89 \pm 1.61}$ | $67.82 \pm 2.30$ |
| | $s = 16$ | $66.83 \pm 5.19$ | $77.42 \pm 3.50$ | $81.93 \pm 2.15$ | $\mathbf{76.12 \pm 5.08}$ | $\mathbf{81.62 \pm 1.87}$ | $82.57 \pm 1.61$ | $\mathbf{68.43 \pm 1.98}$ |
| $K = 1$ | | $67.17 \pm 5.68$ | $76.99 \pm 2.99$ | $81.02 \pm 1.88$ | $\mathbf{75.89 \pm 5.07}$ | $80.92 \pm 1.66$ | $82.40 \pm 1.89$ | $66.95 \pm 1.91$ |
| $K = 2$ | $s = 8$ | $\mathbf{68.00 \pm 5.23}$ | $\mathbf{78.14 \pm 2.81}$ | $82.06 \pm 1.75$ | $75.67 \pm 4.19$ | $\mathbf{81.54 \pm 2.13}$ | $\mathbf{82.89 \pm 1.61}$ | $67.82 \pm 2.30$ |
| $K = 3$ | | $65.75 \pm 5.54$ | $77.75 \pm 2.66$ | $81.85 \pm 1.92$ | $74.96 \pm 5.77$ | $80.82 \pm 1.84$ | $82.53 \pm 1.56$ | $68.35 \pm 2.13$ |
| $K = 4$ | | $65.17 \pm 6.62$ | $77.75 \pm 3.01$ | $\mathbf{82.46 \pm 1.98}$ | $75.09 \pm 4.94$ | $81.17 \pm 2.11$ | $82.26 \pm 1.71$ | $\mathbf{68.35 \pm 1.92}$ |

## 5.2 GRAPH CLASSIFICATION

We further evaluate the different models in graph classification tasks and study the components of BankGCN. Specially, for a fair and complete comparison, we consider two cases, (i) the same number of features, *i.e.*, all of models with the same number of feature maps per hidden layer, and (ii) the same number of parameters, *i.e.*, models with nearly the same number of free parameters per hidden layer, in Table 2 and Table 4 respectively. Experimental settings are presented in App. D.

As presented in Table 2, BankGCN outperforms all the MP baselines with nearly the same or even fewer number of parameters on TU-benchmarks. It further achieves better performance than its counterpart ChebNets with much fewer free parameters, *i.e.*, about $1/(K + 1)$, and is also superior to the most recent spectral method ARMA on most datasets. As presented in Fig. 2, the learned filters have different frequency responses rather than only low-pass, some of them being high-pass and some focusing on middle-frequencies. With such a bank of filters, BankGCN handles the multi-channel signals flexibly and thereby achieves the best performance on almost all the datasets.

We then go one step further to evaluate the adaptive filtering capabilities. As presented in Fig. 2, the learned filters have different frequency responses on various datasets as they are adapted to the data characteristics. As listed in Tables 2 and 3, the BankGCN ($s = 1$) employing one single adaptive filter still outperforms GCN with 'low-pass' filtering on most datasets; Furthermore, BankGCN is superior to its variant BankGCN-Diff that uses predefined filter banks on all the datasets. These validate the benefits of adaptive filtering to flexibly capture the spectral characteristics of data.

**Study on the number of filters.** Furthermore, we evaluate the adoption of filter banks rather than a single filter, and study the impact of the number of filters in the filter bank (equivalently $s$, the number of subspaces) on the classification performance. With a group of filters, the ability of convolution operators to handle information is enhanced. As presented in Table 2, BankGCN-Diff outperforms GCN with additional band-pass and high-pass filters on almost all the datasets, and BankGCN further improves the performance with adaptive filters. Furthermore, as $s$ increases from 1 to 8, the performance of BankGCN is improved on most datasets, as listed in Table 3. These validate the benefits of using more than one filter. With $s$ further increased into 16, the performance is degraded on several datasets. Given that the total dimension of all the subspaces is fixed, the dimension of each space decreases and the representation capacity of each subspace probably declines with the further growth of $s$.

**Study on the order of filters.** The order of polynomials determines the function space of filters. In the graph frequency domain, as demonstrated in Fig. 2, it can better realize the bandpass property

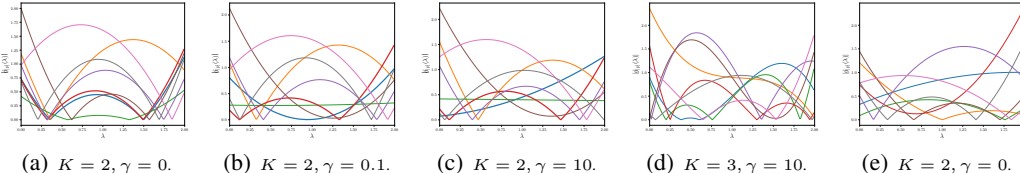

| (a) $K = 2, \gamma = 0.$ | (b) $K = 2, \gamma = 0.1.$ | (c) $K = 2, \gamma = 10.$ | (d) $K = 3, \gamma = 10.$ | (e) $K = 2, \gamma = 0.$ |

Figure 2: Illustrations of the frequency responses of the learned filters of BankGCN ($s = 8$) in the first layer of networks. (a) $\sim$ (d) are on NCI109 and (e) on FRANKENSTEIN.

of filters using a larger $K$ but at the cost of a greater risk of overfitting the spectrum of training data. In the spatial domain, the value of $K$ corresponds to the message passing range, and a large $K$ will affect the locality of signals. Thereby, a tradeoff is needed. As shown in Table 3, BankGCN with $K = 2$ achieves the best performance on most datasets. We notice that for the cases with complex node signals, like FRANKENSTEIN with 780-channel node attributes, a relatively large $K$ is needed to exploit their various spectral characteristics; and the small $K$ is preferred on simple datasets, such as the PROTEINS dataset with 3-channel node category features.

**Study on the regularization.** We further evaluate the effect of the proposed diversity regularization. The comparison of BankGCN-NR and BankGCN in Table 2 shows that the regularization improves the classification performance on almost all the datasets. On the FRANKENSTEIN dataset whose signal is composed of 780-channel attributes, the regularization is not helpful. We infer that the information in such high-channel signal is complex enough to induce different filters, as presented in Fig. 2(e). Fig. 2(a)-(c) show that the learned filters in a filter bank with regularization present better diversity in frequency response, than those without regularization. For example, the filters denoted by blue and red in Fig. 2(a) are with similar frequency responses, while they are more diverse in Fig. 2(b) and Fig. 2(c). This is further verified by the maximum similarity scores of the polynomial coefficients that define the filters $\Omega(\alpha) = 0.997, 0.744,$ and $0.649$ (computed as Eq. 14) for Fig. 2(a)-(c), respectively. More results and analysis are presented in App. E.

BankGCN still achieves the best performance on both CIFAR-10 and Ogbg-molhiv with all the models having a similar number of free parameters per hidden layer, as presented in Table 4. Furthermore, to evaluate the generalization of models, we construct a reduced CIFAR-10 (1000) dataset by taking 100 graphs per category to form the training set, while maintaining the validation and testing sets. BankGCN still performs best on the reduced CIFAR-10 dataset and is among the models with

Table 4: Classification accuracy on CIFAR-10 and Ogbg-molhiv (no edge attributes) datasets.

| Method | CIFAR-10 | CIFAR-10 (1000) | | Ogbg-molhiv |
| | Acc | Acc | Decrease | ROC-AUC |
| --- | --- | --- | --- | --- |
| GCN | $55.64 \pm 0.11$ | $36.47 \pm 0.31$ | -34.5% | $75.18 \pm 1.85$ |
| GraphSage | $63.51 \pm 0.40$ | $40.03 \pm 0.56$ | -37.0% | $75.39 \pm 1.64$ |
| GIN | $50.04 \pm 0.06$ | $31.97 \pm 0.20$ | -36.1% | $71.52 \pm 1.45$ |
| GAT | $60.34 \pm 0.19$ | $36.08 \pm 0.04$ | -40.2% | $75.08 \pm 0.39$ |
| ChebNets ($K = 2$) | $64.33 \pm 0.14$ | $39.46 \pm 0.75$ | -38.7% | $74.69 \pm 2.08$ |
| ChebNets ($K = 3$) | $63.62 \pm 0.23$ | $37.91 \pm 0.40$ | -40.4% | $73.17 \pm 1.57$ |
| ARMA ($K = 2$) | $61.66 \pm 0.35$ | $32.66 \pm 0.09$ | - 47.0% | $75.73 \pm 1.15$ |
| ARMA ($K = 3$) | $61.79 \pm 0.28$ | $32.47 \pm 0.32$ | - 47.5% | $74.61 \pm 0.98$ |
| BankGCN($K = 2, s = 16$) | $\mathbf{66.17 \pm 0.34}$ | $42.82 \pm 0.33$ | -35.3% | $\mathbf{77.95 \pm 1.96}$ |
| BankGCN($K = 3, s = 16$) | $66.00 \pm 0.51$ | $\mathbf{42.95 \pm 0.49}$ | -34.9% | $75.72 \pm 1.45$ |

the least performance loss compared with the full dataset. Together with ROC-AUC being a measure of the generalization ability of a model, BankGCN performs well in the sense of generalization, especially when compared with its counterpart ChebNets. These further validate the effect of the proposed sharing scheme between filters for model simplification on improving generalization.

# 6 CONCLUSION

In this paper, we propose a novel graph convolution operator, termed BankGCN, constructed on an adaptive filter bank for graph representation learning. The filter bank is equivalent to a group of learnable message passing schemes in $K$-hop neighborhoods. Together with subspace decomposition, BankGCN explores a sharing scheme between filters to adaptively handle information of diverse spectral characteristics with significantly fewer parameters than its competitors, and achieves excellent performance on node classification and graph classification tasks. An interesting direction for future research resides in discussing the capacity of the proposed graph convolution operator in terms of graph isomorphism test. It may also be promising to employ BankGCN in a variety of tasks on non-Euclidean data like 3-D point cloud classification and segmentation.

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

## A  NOTATIONS AND PRELIMINARIES

In this paper, we consider data that is represented on undirected graphs. An input graph is described as $\mathcal{G} = (\mathcal{V}, \mathcal{E})$, with $\mathcal{V}$ and $\mathcal{E}$ respectively denoting the set of vertices and the set of edges. We generally use capital letters for matrices and bold lowercase letters for vectors. The graph topology is characterized by the adjacency matrix $A$ with a non-zero value $(A)_{ij}$ indicating the weight of the edge connecting vertices $v_i$ and $v_j$. If available [3], the $d$-channel signals or attributes of nodes are represented as $\boldsymbol{x}(v_m) \in \mathbb{R}^d$ for $\forall v_m \in \mathcal{V}$, and $X = [\boldsymbol{x}(v_1), \boldsymbol{x}(v_2), \cdots, \boldsymbol{x}(v_n)]^T$ is for the whole graph $\mathcal{G}$ with $n = |\mathcal{V}|$ nodes. Specially, for vector features, $x_i(v_m)$ indicates the $i$-th channel of the signal on vertex $v_m$, and $\boldsymbol{x_i} = [x_i(v_1), x_i(v_2), \cdots, x_i(v_n)]^T$ represents this signal on the whole graph. For the hierarchical structure of GCNs, we employ a subscript $l$ to indicate variables or parameters belonging to the $l$-th layer. In most cases, we ignore $l$ for clarity without causing confusion.

The Graph Fourier Transform (GFT) is defined based on the graph Laplacian matrix $L$ (Shuman et al., 2013). We adopt the symmetric normalized graph Laplacian operator, i.e., $L = I - D^{-\frac{1}{2}} A D^{-\frac{1}{2}}$ with the diagonal degree matrix $D$ defined as $(D)_{ii} = \sum_j (A)_{ij}$. The eigen-decomposion of the Laplacian matrix is denoted by $L = U \Lambda U^*$, where $U = [\boldsymbol{u_1}, \boldsymbol{u_2}, \ldots, \boldsymbol{u_n}]$ is composed of the eigenvectors $\boldsymbol{u_k}$, $k = 1, \cdots, n$ and $\Lambda$ is a diagonal matrix with eigenvalues $\lambda_1, \lambda_2, \ldots, \lambda_n$, with $0 = \lambda_1 \leq \lambda_2 \leq \cdots \leq \lambda_n \leq 2$. The eigenvalues $\{\lambda_k\}$ construct the spectrum

---

[3]If node attributes are not available, we adopt the structural information like the one-hot encoding representation of node degree and the local clustering coefficient as node signals.

of the graph $\mathcal{G}$ [4], and the GFT of a signal $\boldsymbol{x}$ is calculated as the inner-product between each of its component $\boldsymbol{x_i}$ and the eigenvectors $\{\boldsymbol{u_k}\}$ (Shuman et al., 2013):

$$\hat{x}_i(\lambda_k) = \sum_{m=1}^{n} x_i(v_m)u_k^*(v_m), \tag{S1}$$

where $*$ indicates conjugate transpose and $u_k^*(v_m)$ denotes the $m$-th element in $\boldsymbol{u_k^*}$. The Inverse Graph Fourier Transform (IGFT) is defined as

$$x_i(v_m) = \sum_{k=1}^{n} \hat{x}_i(\lambda_k)u_k(v_m). \tag{S2}$$

According to the convolution theorem, the convolution between the $i$-th channel of the signal $x_i(v_m)$ and the corresponding filter $g(v_m)$ (with frequency response $\hat{g}(\lambda_k)$ ) is defined as:

$$(x_i * g)(v_m) = \sum_{k=1}^{n} \hat{x}_i(\lambda_k)\hat{g}(\lambda_k)u_k(v_m) \tag{S3}$$

Let us define $\hat{g}(\Lambda) = \mathrm{diag}([\hat{g}(\lambda_1), \hat{g}(\lambda_2), \ldots, \hat{g}(\lambda_n)])$. We have for $\boldsymbol{x_i}(v_m)$ that

$$(\boldsymbol{x_i} * g) = U\hat{g}(\Lambda)U^*\boldsymbol{x_i}. \tag{S4}$$

Finally, a graph filter bank is composed of a set of filters $\{\hat{g}_i(\Lambda)\}$ to decompose a graph signal into a series of signals with different frequency components (Narang & Ortega, 2012).

## B  PROOF OF PROPOSITION 1

*Proof.* For spectral methods with polynomial filters $\hat{g}(\lambda) = \sum_{k=0}^{K} \theta^{(k)}T_k(\lambda)$, the $j$-th channel ($j = 1, 2, \ldots, d'$) of the output signal is

$$\begin{aligned}
\boldsymbol{h_j} &= \sum_{i=1}^{d} g_{ij} * \boldsymbol{x_i} = \sum_{i=1}^{d} U\hat{g}_{ij}(\Lambda)U^*\boldsymbol{x_i} \\
&= \sum_{i=1}^{d} U \sum_{k=0}^{K} \Theta_{i,j}^{(k)}T_k(\Lambda)U^*\boldsymbol{x_i} = U \sum_{i=1}^{d} \sum_{k=0}^{K} \Theta_{i,j}^{(k)}T_k(\Lambda)\hat{\boldsymbol{x_i}}.
\end{aligned} \tag{S5}$$

Thereby, the spectrum of output signal is a linear transform of the $K$-order polynomial expansion of input signals $\{T_k(\Lambda)\hat{\boldsymbol{x_i}}\}_{i,k}$

$$\hat{\boldsymbol{h_j}} = \sum_{k=0}^{K} \sum_{i=1}^{d} \Theta_{i,j}^{(k)}T_k(\Lambda)\hat{\boldsymbol{x_i}}. \tag{S6}$$

For BankGCN, we assume that the $j$-th channel ($j = 1, 2, \ldots, d'$) of the output signal corresponds to the $q$-th channel of the filtered signal in the $p$-th subspace, and we have

$$\begin{aligned}
\boldsymbol{h_j} &= (g_{[p]} * \boldsymbol{r_{[p]q}}) = (g_{[p]} * (\sum_{i=1}^{d}(W_{[p]})_{i,q}\boldsymbol{x_i} + b_{[p]q}\mathbf{1})) \\
&= U(\sum_{k=0}^{K} \alpha_{[p]}^{(k)} \sum_{i=1}^{d}(W_{[p]})_{i,q}T_k(\Lambda)U^*\boldsymbol{x_i} + \boldsymbol{\epsilon_{[p]q}}),
\end{aligned} \tag{S7}$$

where $\boldsymbol{r_{[p]q}}$ denotes the $q$-th channel of the projected signal $R_{[p]}$ in the $p$-th subspace, $b_{[p]q}$ indicates the $q$-th element in $\boldsymbol{b_{[p]}}$, $\mathbf{1}$ denotes a constant vector with value 1, and $\boldsymbol{\epsilon_{[p]q}}$ represents a vector independent on input signals. Correspondingly, the spectrum of $\boldsymbol{h_j}$ is

$$\hat{\boldsymbol{h_j}} = \sum_{k=0}^{K} \sum_{i=1}^{d} \alpha_{[p]}^{(k)}(W_{[p]})_{i,q}T_k(\Lambda)\hat{\boldsymbol{x_i}} + \boldsymbol{\epsilon_{[p]q}}. \tag{S8}$$

According to Eq. S6 and Eq. S8, BankGCN can capture and preserve diverse frequency components of input graph signals, like its counterpart introduced in Eq. 2                                     □

---

[4]In analogy to classical Fourier analysis, eigenvalues provide a corresponding notion of frequency and lead to frequency filtering.

## C  EXPERIMENTAL SETTINGS ON NODE CLASSIFICATION

We evaluate the models on four heterophilic graph datasets (Pei et al., 2020; Tang et al., 2009), including Texas, Wisconsin, Cornell, and Actor. Texas, Wisconsin, Cornell are three subdatasets from WebKB. In these datasets, nodes represent web pages with their 1703-dimensional bag-of-words representation as node features, and hyperlinks between the web pages form edges. Actor is a dataset preprocessed by Pei et al. (2020) from the film-director-actor-writer network (Tang et al., 2009). The graph in the dataset represents the co-occurrence (connection) of actors (nodes) on the same Wikipedia page. There are five categories on all these datasets.

We follow the data split and evaluation procedure of previous methods (Pei et al., 2020; Zhu et al., 2020; Li et al., 2021). Specifically, 48%:32%:20% of nodes per class are split into training, validation and test sets, and 10 such random splits on each dataset are provided by Pei et al. (2020); Fey & Lenssen (2019). The network architecture consists of two layers. We select the following hyper-parameters through grid-search: number of feature maps of the hidden layer $\in \{32, 64, 128\}$, learning rate$\in \{0.02, 0.04\}$, weight decay $\in \{5e^{-4}, 1e^{-3}\}$, and $s \in \{4, 8, 16\}$. In addition, the dropout is $0.5$ only in the hidden layer, and $s$ is set as $1$ for the output layer.

We compare BankGCN with several state-of-the-art graph convolution methods. For the MPGCNs, we consider GCN (Kipf & Welling, 2017), GraphSage (Hamilton et al., 2017), GAT (Velikovi et al., 2018), MixHop (Abu-El-Haija et al., 2019), SGC (Wu et al., 2019), GEOM-GCN (Pei et al., 2020), H2GCN (Zhu et al., 2020), SGF (NT et al., 2021), and ASGAT (Li et al., 2021). Regarding spectral GCNs, we compare with its counterpart ChebNets (Defferrard et al., 2016) that also adopts Chebyshev polynomial filters, and the most recent spectral method ARMA (Bianchi et al., 2021).

## D  EXPERIMENTAL SETTINGS ON GRAPH CLASSIFICATION

**Datasets and data splits.** For TU datasets (Kersting et al., 2016), we conduct experiments on seven widely used public benchmark graph classification datasets, including ENZYMES, D&D, PROTEINS, NCI1, NCI109, MUTAGENICITY, and FRANKENSTEIN [5]. We adopt node categorical features (one-hot encoding) and node attributes as node signals, depending on availability on the datasets, which are usually heterogeneous in different dimensions. Specifically, node attributes are adopted on FRANKENSTEIN, node categorical features and node attributes (normalized in range $[0, 1]$) on ENZYMES, and node categorical features on the other datasets. The statistics and properties of the datasets are summarized in Table S1. According to (Lee et al., 2019), we use stratified sampling to randomly split each dataset into training, validation and test sets with a ratio of 8:1:1. The trained model with the best validation performance is selected for test. In order to alleviate the impact of data partition and network initialization, we conduct 20 random runs with different data splits and network initializations on each dataset, and report the mean accuracy with standard deviation of these 20 test results.

We further adopt two large benchmark datasets, CIFAR-10 (Dwivedi et al., 2020) and Ogbg-molhiv (Hu et al., 2020), in the experiments. The graph version of CIFAR-10 is composed from the superpixels of images, and RGB intensities and normalized coordinates form node signals. Ogbg-molhiv is a molecule graph dataset, with 9-dimensional node features including atomic number, chirality, and additional atom features. We divide data on these two datasets in accordance with (Dwivedi et al., 2020) and (Hu et al., 2020), respectively. Similarly, results are achieved with 3 runs on CIFAR-10 and 10 runs on Ogbg-molhiv in order to alleviate the impact of network initialization.

**Network architectures.** In the experiments, we adopt a similar architecture to (Xu et al., 2018), and the network consists of four convolution layers, one graph-level readout module and one prediction module for all the datasets. Specifically, the graph convolution layer is designed as introduced in Section 4 or defined by various baseline models. An $l_2$ normalization function is utilized in each convolution layer to stabilize and accelerate the training process, as in (Hamilton et al., 2017). A graph-level readout module then aggregates the graph features from all the convolution layers to generate the graph representation $h_{\mathcal{G}}$. It consists of node-wise mean and maximum operators (or mean operator on Ogbg-molhiv as in (Hu et al., 2020)), represented by $\omega(\cdot)$.

$$h_{\mathcal{G}} = \text{Concat}(\omega(X_l)|l = 1, 2, 3, 4). \tag{S9}$$

---

[5] Datasets could be downloaded from https://ls11-www.cs.tu-dortmund.de/staff/morris/graphkerneldatasets

Table S1: Dataset statistics and properties (L indicates node categorical features and A denotes node attributes).

| | ENZ | D&D | PROT | NCI1 | NCI109 | MUTA | FRAN | CIFAR-10 | Ogbg-molhiv |
|---|---|---|---|---|---|---|---|---|---|
| Avg $|\mathcal{V}|$ | 32.63 | 284.32 | 39.06 | 29.87 | 29.68 | 30.32 | 16.90 | 117.63 | 25.51 |
| Avg $|\mathcal{E}|$ | 62.14 | 715.66 | 72.82 | 32.30 | 32.13 | 30.77 | 17.88 | 564.86 | 27.47 |
| Node feature | L+A | L | L | L | L | L | A | A | A |
| Dim(feat) | 3+18 | 89 | 3 | 37 | 38 | 14 | 780 | 5 | 9 |
| #Classes | 6 | 2 | 2 | 2 | 2 | 2 | 2 | 10 | 2 |
| #Graphs | 600 | 1,178 | 1,113 | 4,110 | 4,127 | 4,337 | 4,337 | 60,000 | 41,127 |

Finally, a prediction module composed of a linear fully connected layer and a softmax layer makes the prediction of the category of the input graph. The cross-entropy (represented as $\mathscr{T}_\Theta(\mathcal{G}, Y)$) is adopted as the loss function, which together with the regularization $\Omega(\alpha)$ for diversity condition composes the objective function:

$$\mathscr{T}_\Theta(\mathcal{G}, Y) + \gamma\, \Omega(\alpha) \tag{S10}$$

where $\Theta$ denotes all the free parameters and $Y$ indicates ground truth categorical labels.

In more details, we consider two versions of the architectures in the experiments in order to evaluate the models under the same number of features and the same number of parameters, respectively. For the first case, all of models have the same number of feature maps per hidden layer, whereas models are composed of nearly the same number of parameters per hidden layer for the latter case. We consider the first version with TU datasets in Table 2 with 64 feature maps per hidden layer, and the second version in Table 4 with nearly 16,500 and 65,800 free parameters per layer for all the models on CIFAR-10 and Ogbg-molhiv, respectively.

**Configurations.** We implement the proposed models in Pytorch (Paszke et al., 2017) with geometric package (Fey & Lenssen, 2019), and optimize all of the models with the Adam optimizer (Kingma & Ba, 2015) on workstations with GPU GeForce GTX 1080 Ti for TU datasets as well as CIFAR-10 and with GPU RTX 2080 Ti for Ogbg-molhiv. In more details, for TU datasets, the learning rate is 0.001 and the batch size is 64. The number of training epochs is set as 500, and early stopping is employed with patience 30. Finally, we obtain the following optimal hyper-parameters through grid search: weight decay $\in \{0, 1e^{-5}, 1e^{-4}\}$ and $\gamma \in \{0, 0.1, 10\}$.

For CIFAR-10 and Ogbg-molhiv, the batch size is increased to 256 due to their large scale. Similarly to (Dwivedi et al., 2020), we adopt a dynamic learning rate that is initialized as 0.001 and decays by 0.1 when validation loss is not improved for 20 epochs until the minimum learning rate $1e^{-5}$. The number of training epochs is 500 with early stoping (patience 50). The other settings are the same as TU datasets.

For the polynomial basis in the filter design of BankGCN, we adopt the widely used Chebyshev polynomials.

$$T_0(\lambda) = 1,\; T_1(\lambda) = \lambda,\; T_k(\lambda) = 2\lambda T_{k-1} - T_{k-2}. \tag{S11}$$

The graph Laplacian is adopted as $\tilde{L} = L - I$ for numerical stability, similarly to (Defferrard et al., 2016).

**Baselines.** We compare BankGCN with several state-of-the-art graph convolution methods. For the MPGCNs, we consider GCN (Kipf & Welling, 2017), GraphSage (Hamilton et al., 2017) with mean aggregation, GAT (8-heads) (Velikovi et al., 2018), and GIN using SUM-MLP (2 layers) that achieves the best performance (Xu et al., 2019). Regarding spectral GCNs, we compare with its counterpart ChebNets (Defferrard et al., 2016) that also adopts Chebyshev polynomial filters, and the most recent spectral method ARMA (1-stack) (Bianchi et al., 2021). For a fair comparison, the results of baseline models are obtained with the same configurations as BankGCN using the public versions provided in the pytorch-geometric package (Fey & Lenssen, 2019), with their respective optimal hyper-parameters (weight decay) selected by grid search.

**Ablation Study.** Furthermore, we consider two simplifications of the proposed model for ablation study. First, **BankGCN-Diff** adopts a predefined filter bank. The filter bank consists of a low-pass filter, a high-pass filter, and band-pass filters from graph diffusion wavelets as used by Gama et al. (2019); Min et al. (2020). The specific frequency responses of these filters are presented in

Table S2: Ablation study on the diversity regularization with BankGCN ($K = 2$, $s = 8$).

| | ENZY | DD | NCI1 | PROT | NCI109 | MUTA | FRAN |
|---|---|---|---|---|---|---|---|
| $\gamma = 0$ | $65.83 \pm 6.66$ | $77.03 \pm 4.08$ | $81.89 \pm 1.95$ | $75.36 \pm 4.68$ | $81.03 \pm 1.95$ | $82.44 \pm 1.69$ | $\mathbf{67.82 \pm 2.30}$ |
| $\gamma = 0.01$ | $66.50 \pm 6.39$ | $77.54 \pm 3.21$ | $81.63 \pm 2.33$ | $75.49 \pm 4.07$ | $81.27 \pm 2.16$ | $82.87 \pm 1.94$ | $67.71 \pm 1.96$ |
| $\gamma = 0.1$ | $\mathbf{68.00 \pm 5.23}$ | $78.09 \pm 2.18$ | $\mathbf{82.06 \pm 1.75}$ | $75.67 \pm 4.19$ | $81.34 \pm 1.92$ | $82.83 \pm 1.87$ | $67.68 \pm 2.10$ |
| $\gamma = 1$ | $66.42 \pm 6.20$ | $77.97 \pm 3.57$ | $81.81 \pm 1.97$ | $\mathbf{76.29 \pm 4.84}$ | $81.00 \pm 2.21$ | $\mathbf{82.95 \pm 1.44}$ | $67.76 \pm 1.65$ |
| $\gamma = 10$ | $66.75 \pm 5.90$ | $\mathbf{78.14 \pm 2.81}$ | $82.06 \pm 2.10$ | $75.31 \pm 4.64$ | $\mathbf{81.54 \pm 2.13}$ | $82.89 \pm 1.61$ | $67.20 \pm 1.67$ |
| $\gamma = 100$ | $67.08 \pm 5.42$ | $77.75 \pm 3.26$ | $81.81 \pm 2.18$ | $75.98 \pm 4.40$ | $81.14 \pm 1.94$ | $82.44 \pm 1.71$ | $67.52 \pm 1.83$ |

Appendix D. Second, **BankGCN-NR** removes the diversity regularization from BankGCN with $\gamma = 0$.

**BankGCN-Diff** adopts a predefined filter bank that consists of a low-pass filter, a high-pass filter, and band-pass filters from graph diffusion wavelets as used in (Gama et al., 2019; Min et al., 2020). The specific frequency responses of these filters are

$$\psi_{l,0}(T) = I - T, \qquad\qquad \psi_{h,0}(T) = I - T'$$
$$\psi_{l,1}(T) = T(I - T), \qquad\qquad \psi_{h,1}(T) = T'(I - T')$$
$$\psi_{l,2}(T) = T^2(I - T^2), \qquad\qquad \psi_{h,2}(T) = T'^2(I - T'^2)$$
$$\psi_{l,3}(T) = T^4(I - T^4), \qquad\qquad \psi_{h,3}(T) = T'^4(I - T'^4) \qquad\qquad \text{(S12)}$$

with two diffusion operators:

$$T = (1 - D^{-\frac{1}{2}}AD^{-\frac{1}{2}})/2, \quad T' = (1 + D^{-\frac{1}{2}}AD^{-\frac{1}{2}})/2. \qquad\qquad \text{(S13)}$$

Their frequency responses can be easily derived with the relationship between the diffusion operators and the graph Laplacian. Specifically, $\psi_{l,0}$ and $\psi_{h,0}$ are respective low-pass and high-pass filters, and the others are band-pass filters.

## E    MORE EXPERIMENTAL RESULTS AND ANALYSIS

**More Ablation Results about the Diversity Regularization.** In Table S2, we consider the values of $\gamma \in \{0, 0.01, 0.1, 1, 10, 100\}$ to adjust its contribution in the objective function (Eq. 16). The regularization improves the classification performance on almost all the datasets. Mostly, the best performance is achieved with $\gamma = 0.1$ or $\gamma = 10$, and thereby we consider $\gamma \in \{0, 0.1, 10\}$ in adjusting the contribution of regularization in the experiments. As illustrated in Fig. S1(a)-(f), the learned filters in a filter bank with regularization present better diversity in terms of frequency response than those without regularization on NCI109 and ENZYMES datasets. For example, the filters denoted by blue and red in Fig. S1(a) are with similar frequency responses, while they are more diverse in Fig. S1(b) and Fig. S1(c). Besides, the filters indicated by gray and purple in Fig. S1(d)-(f) show similar results. These are further verified with the maximum similarity scores of the polynomial coefficients that define the filters $\Omega(\alpha) = 0.997, 0.744$, and $0.649$ (computed as Eq. (18)) for Fig. S1(a)-(c), and $\Omega(\alpha) = 0.899, 0.714$, and $0.705$ for Fig. S1(d)-(f), respectively.

**More illustration results.** We provide more illustration results in Fig. S2 to demonstrate adaptive filter banks with different numbers of filters. With the increase of the $s$, BankGCN is more flexible to decompose the signal into components with different spectral characteristics. Furthermore, as shown in Fig. S3, the frequency responses of the learned filters in a filter bank vary with the datasets and the layers in the neural networks. These demonstrate the necessity and benefit of adopting learnable rather than predefined filters.

## F    MORE COMPARISONS WITH RELATED WORKS

**Message Passing Graph Convolutional Networks.** MPGCNs design graph convolution with a variety of message passing schemes in the spatial domain. For instance, messages are aggregated with the node-wise mean or max operation in a localized neighborhood in GraphSage (Hamilton et al., 2017), or based on attention scores in GAT (Velikovi et al., 2018). A more expressive scheme, GIN,

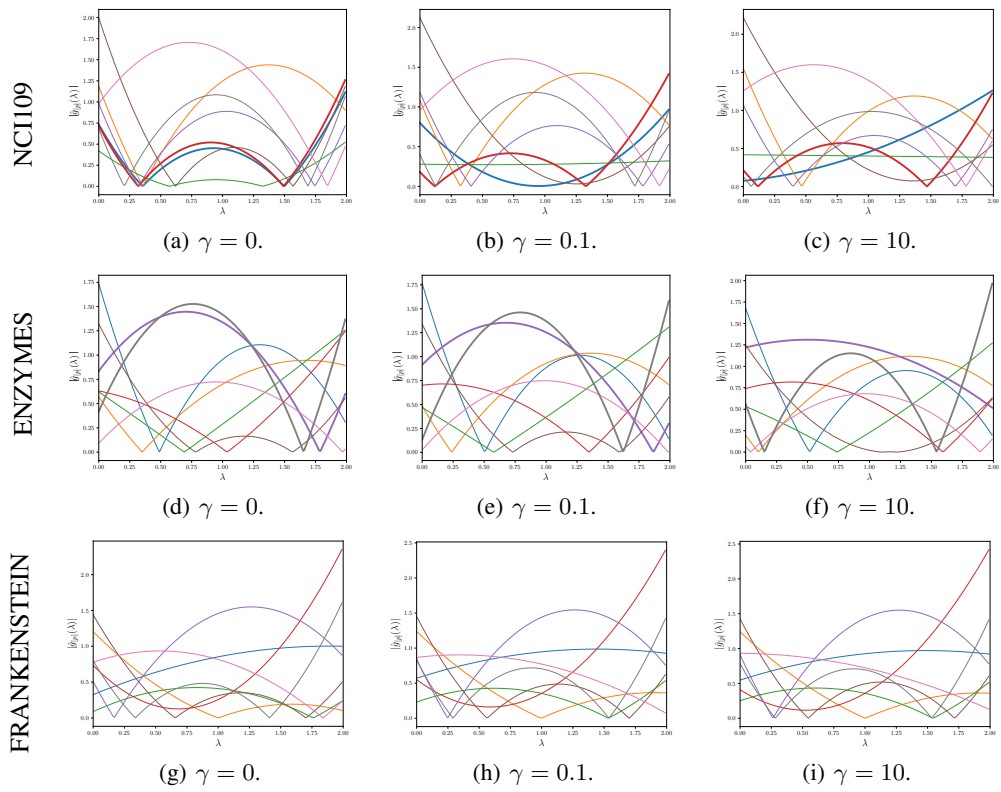

Figure S1: Comparisons of the frequency responses of the learned filters of BankGCN ($K = 2, s = 8$) under different regularization strengths on different datasets.

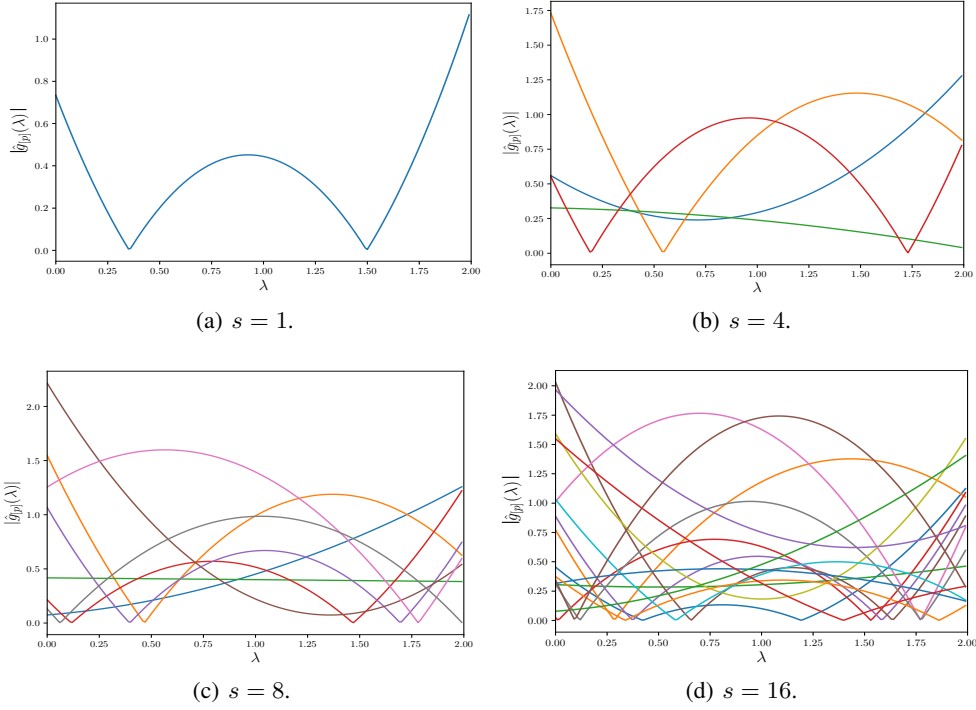

Figure S2: Illustrations of the frequency responses of the learned filters of BankGCN ($K = 2, \gamma = 10$) in the first layer of networks with different number of subspaces ($s$) on NCI109.

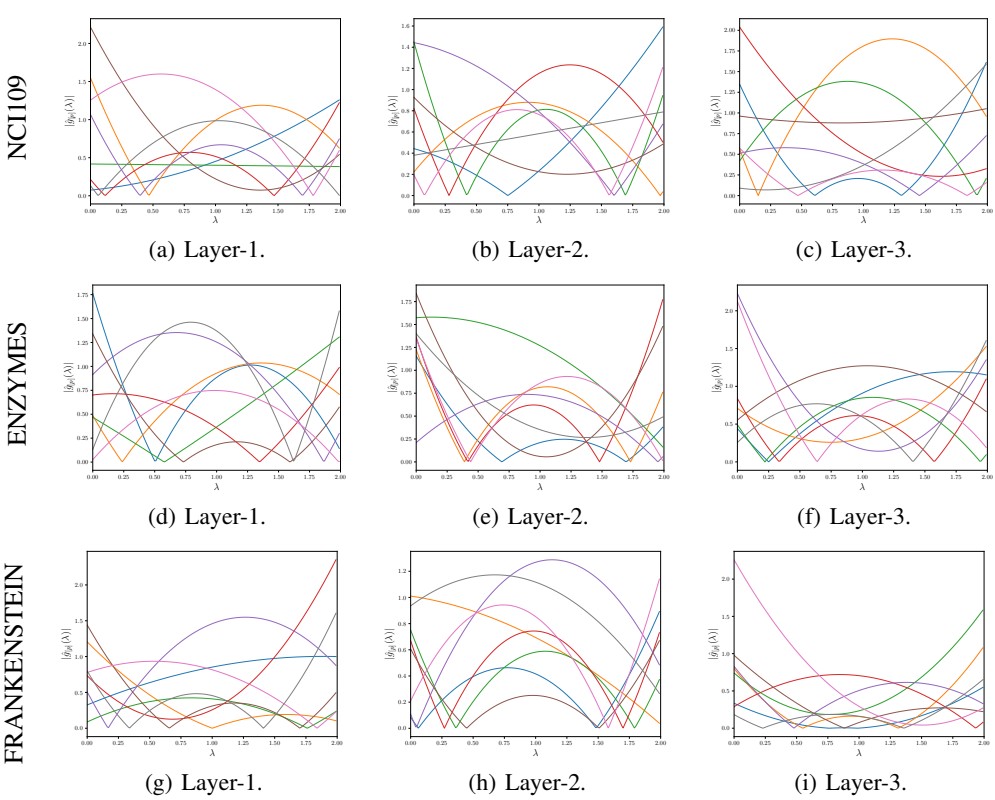

Figure S3: Illustrations of the frequency responses of the learned filters of BankGCN ($K = 2, s = 8$) in the different layers of networks on different datasets.

is further proposed with summing features in a neighborhood followed by a multi-layer perceptrons (MLP) to approximate any injective function on multiset (Xu et al., 2019). However, these methods are typically constrained to a single one-hop message passing strategy that mostly captures 'low-pass' characteristics in the graph data. Klicpera et al. (2019) expand the MP range to multi-hop neighborhoods with graph diffusion operators. Besides, several methods are proposed to implement a high-order (polynomial) filter with the whole network. For example, SGC (Wu et al., 2019) corresponds to a fixed polynomial filter, and LGC (Navarin et al., 2020), GPR-GNN (Chien et al., 2021), SGF (NT et al., 2021), and S$^2$GC (Zhu & Koniusz, 2021) further realizes an adaptive filter with diverse motivations. Alternatively, a group of methods attempt to complement the 'low-pass' features of GCN using different strategies. For instance, Scattering GCN (Min et al., 2020) resorts to geometric scattering transform (Gao et al., 2019) to complement the 'low-pass' features of GCN (Kipf & Welling, 2017) with 'band-pass' features, a one-order high-pass filter is adopted as the complement (Bo et al., 2021), and SpGAT (Chang et al., 2021) employs predefined graph wavelet filters. Besides, MixHop (Abu-El-Haija et al., 2019) explores linear mixing of neighborhood information to further realize difference operators, which can capture high-frequency information, and H2GCN (Zhu et al., 2020) additionaly aggregates information from high-order neighborhoods. Contrary to these methods, we employ a learnable filter bank with various frequency responses to adaptively capture diverse spectral characteristics of signals rather than merely adding 'high-pass' features or specific components via predefined graph wavelets. Finally, some works further consider adaptive filters (Li et al., 2021; Dong et al., 2021; Pasa et al., 2021), but BankGCN is a more powerful and compact design with polynomial filter banks to avoid eigen-decomposition and a further consideration of the sharing scheme between filters to reduce free parameters. For instance, AdaGNN (Dong et al., 2021) implements an equivalent filter for each channel of input signals through the whole network, while BankGCN can realize it within a single convolution operator. Furthermore, the filter space of BankGCN with $K$-order polynomial filters is larger than AdaGNN, since the $K$-order filter in AdaGNN is obtained by multiplying $K$ one-order filters. The expressive capacity can be further enhanced by stacking multiple BankGCN layers.

