# OpenReview forum: "Graph Convolutional Networks via Adaptive Filter Banks"
_ICLR.cc/2022/Conference — ICLR 2022 Submitted_

### Official Review · Reviewer_84WF · 2021-10-30

**Correctness:** 4
**Technical Novelty And Significance:** 3
**Empirical Novelty And Significance:** 2
**Recommendation:** 6
**Confidence:** 4

**Main Review:**

Strengths:
Graph neural networks are timely and of great interest to researchers.

Weaknesses:

Integrating filter banks in graph neural networks is not new. There has been many studies, such as the work of Tremblay et al. (design of graph filters and filterbanks) and Bianchi et al. (ARMA-based filtering in graph neural networks), as well as related work like Thanou et al. (parametric dictionary learning for signals on graphs). Besides these cited papers, there are other papers on the same topic (in filter-adaptation for graph convolutional networks), such as:
- Li, S., Kim, D., & Wang, Q. (2021). Beyond Low-Pass Filters: Adaptive Feature Propagation on Graphs. ECML/PKDD.
- Wu, Z., Pan, S., Long, G., Jiang, J., & Zhang, C. (2021). Beyond Low-pass Filtering: Graph Convolutional Networks with Automatic Filtering. ArXiv, abs/2107.04755.
- Bo, D., Wang, X., Shi, C., & Shen, H. (2021). Beyond Low-frequency Information in Graph Convolutional Networks. In Proceedings of the AAAI Conference on Artificial Intelligence (Vol. 35, No. 5, pp. 3950-3957).
Only the latter was cited in the submitted paper. While these papers are new, they do cite older papers on the same topic as this work.

Roughly, the proposed method consists of integrating FIR filter banks, with the ARMA-based filtering of Bianchi et al. considers IIR filter banks.

Experiments need to be extensive. While the authors consider several conventional graph convolutional networks in the comparative analysis, only the ARMA-based filtering of Bianchi et al. seems to be comparable in terms of adaptive filter banks. The authors need to provide extensive experiments by comparing to other related methods, considering at least the aforementioned ones.

The authors need to provide a thorough analysis of the computational complexity of the proposed method. While there is one sentence on computational complexity "It achieves linear computational complexity with O(K|E|d) and constant learning complexity, similarly to most existing MPGCNs", this needs to be demonstrated. In practice, the proposed method requires including a filter bank per layer and estimating its parameters can be cumbersome for large-scale graphs and multiple channels. The authors need to provide some experimental analysis of both the memory usage and the computational time, and comparing them to other methods in the literature.
-------------------------
We thank the authors for addressing our concerns.

**Summary Of The Paper:**

This paper proposes a graph convolutional networks where, at each layer, the graph signal is decomposed with an adaptive filter bank.

**Summary Of The Review:**

Interesting work. However, there are several related methods in the literature.

---

### Official Review · Reviewer_rmns · 2021-11-01

**Correctness:** 3
**Technical Novelty And Significance:** 2
**Empirical Novelty And Significance:** 3
**Recommendation:** 3
**Confidence:** 5

**Main Review:**

Strengths:
- The paper is easy to follow with clear motivation and is well written. The design of BankGCN is closely rooted in Graph Signal Processing, which has merits.
- Due to the common assumption on the homophily of underlying graphs, it is of great importance to go beyond low-pass filter types of GNNs.
- Empirical observations on the filter bank and its response to the proposed model are interesting.

Weaknesses:
- The novelty of the proposed BankGCN is limited. BankGCN is closely related to polynomial GNNs with regularized and learnable coefficients. Thus, several designs towards simplifying polynomial GNNs that were proposed recently [1,2,3] might have relatively the same power of expressiveness, since the learned filters could be very similar. The authors are suggested to mention them, add discussions, and especially have an empirical comparison with them to demonstrate the superiority of BankGCN. Meanwhile, several other attempts that try to integrate adaptive frequency response filters into the design of GCNs such as [4,5] are missed and should be added in the discussion of Related Work.
- As one main highlight of the design of BankGCN is to use the filter of K-order polynomial function space instead of computational-intense eigendecomposition, this sacrifice should be discussed more in detail, such as the comparison of the flexibility of learned filters and the degree of acceleration through computational analysis.
- For experiments, the concerns include:
    - The diversity spectral information of the chosen benchmark datasets, such as the average homophily ratio of all graphs, should be quantitatively analyzed since this is the main contribution of the proposed model, and how the learned filters correspond to specific graphs would demonstrate more convincingly.
    - Most graphs in the benchmarks seem to be small and it is a known problem with those small and noisy datasets. Thus, instead of simply averaging several runs, cross-validation should be conducted and report the confidence interval.
    - More recent baselines that also aim to use K-order polynomial function to go beyond low-frequency in graphs should be added. I understand that the design of BankGCN is inspired by the graph signal processing, but the proposed architecture is so similar to these methods, and only comparing with spectral-based GNNs is not adequate.

[1] Polynomial Graph Convolutional Networks, OpenReview 2020

[2] Stacked Graph Filter, arXiv 2021

[3] Simple Spectral Graph Convolution, ICLR 2021

[4] Graph Neural Networks with Adaptive Frequency Response Filter, CIKM 2021

[5] Spectral Graph Attention Network with Fast Eigen-approximation, CIKM 2021


**Summary Of The Paper:**

This paper proposes a new variant of GCN model, namely BankGCN, based on a novel graph convolutional operator. The main idea of BankGCN is to the sharing scheme among different filters to adaptively capture information from different frequencies. Through detailed discussion, BankGCN is claimed to be equivalent to the learnable message passing mechanism across K-hop neighborhood. Experiments on graph classification tasks over a collection of graph benchmark datasets demonstrate that BankGCN could outperform many state-of-the-art spectral-based GNNs.

**Summary Of The Review:**

My concorns are mainly from two aspects:
- The novelty of the proposed method is limited. The contribution of this paper needs further discussion between BankGCN and other efforts on designing filters from K-order polynomial function space.
- The current experimental analysis is insufficient and can be further improved to make it more convincing.

---

> ### Author Response · Authors · 2021-11-23
> **Responses to Reviewer rmns:**
>
> We would like to thank you for your valuable comments.
>
> 1. Discussion and comparison with some works on simplifying polynomial GNNs.
>
>     [A] The whole networks of SGF (NT et al., 2021) and S^2GC (Zhu & Koniusz, 2021) correspond to a high-order polynomial filter, whereas BankGCN, as a convolution operator, is equivalent to a filter bank composed of multiple such polynomial filters with diverse frequency responses. Thereby, BankGCN is more powerful than SGF and S^2GC in terms of graph filtering. Furthermore, compared with PGC (Pasa et al., 2021), BankGCN implements a filter bank with multiple K-order polynomial filters with much fewer parameters, using the proposed sharing scheme between filters and subspace projection. Specifically, the number of free parameters is nearly independent of the polynomial order K, while that of PGC is linear with K, similarly to existing spectral GCNs. Thereby, BankGCN is expected to have better generalization. We have provided this discussion in Related Work (Section 2 and App. F) in the revised version. We have further compared with SGF, adaptive schemes such as ASGAT (Li et al., 2021), and other related works in node classification tasks, and BankGCN achieves the best performance, as presented in Table 1. More experimental comparison and analysis are presented in Section 5.1 in the revised version.
>
>
> 2. Discussion with works on adaptive frequency response filters into the design of GCNs in Related Work.
>
>    [A] We have discussed the suggested references, AdaGNN (Dong et al., 2021) and SpGAT (Chang et al., 2021), in Related Work (Section 2 and App. F) in the revised version. Specifically, AdaGNN implements an equivalent filter for each channel of input signals through the whole network, while BankGCN can realize it within a single convolution operator. Furthermore, the filter space of BankGCN with K-order polynomial filters is larger than AdaGNN, since the K-order filter in AdaGNN is obtained by multiplying K one-order filters. The expressive capacity can be further enhanced by stacking multiple BankGCN layers. On the other hand, SpGAT employs predefined wavelet filters and only considers two filters in a common space per layer, while BankGCN adopts adaptive filter banks together with subspace projection to flexibly decompose and capture the diverse frequency characteristics of graph signals. These comparisons and contributions presented here and in the preceding answers demonstrate the novelty of our work.
>
>
> 3. Strengthen the advantage of the design of BankGCN using polynomial filters instead of computational-intense eigendecomposition.
>
>     [A] Thanks for the advice. We have strengthened the advantages of adopting polynomial filters to avoid computational-intense eigendecomposition. Given that the detailed comparison between polynomial filters and eigendecomposition-based methods have been discussed and analyzed in ChebNets (Defferrard et al., 2016), we do not elaborate on it due to space limitation, since the proposed BankGCN as a further evolution of ChebNets would naturally inherit of such merits.
>
>
> 4.  Concerns of experiments
>
>      4.1 The average homophily ratio of all graphs.
>
>     [A] We have measured the homophily ratio in node classification tasks, as shown in Table 1.
>
>     4.2 Cross-validation should be conducted and report the confidence interval.
>
>     [A] The mean and standard variation of results of 20 runs adopted in the experiments can also alleviate the impact of randomness, consistently with previous methods like (Velikovi et al., 2018).
>
>     4.3 Adding more recent baselines also using K-order polynomial function to go beyond low-frequency in graphs.
>
>     [A] We have further added and discussed more recent baselines using K-order polynomial functions in Related Work in the revised version (Section 2 and App. F). Furthermore, we have compared BankGCN with some of them in additional experiments in node classification tasks. As shown in Table 1, BankGCN outperforms all the baselines, including adaptive filter based methods such as SGF and ASGAT. More results are presented in Section 5.1.

---

### Official Review · Reviewer_nrop · 2021-11-02

**Correctness:** 3
**Technical Novelty And Significance:** 2
**Empirical Novelty And Significance:** 3
**Recommendation:** 5
**Confidence:** 3

**Main Review:**

**Strengths**

(1) This paper addresses the limited flexibility of message passing convolutional networks (MPGCNs) because of their low-pass properties, which are one of the most important issues of Graph Neural Networks.

(2) This paper is well written and structured.

(3) This paper provides comprehensive experiments. In particular, the discussion of the number/order of filters and regularization is well presented to understand.

**Weakness**

(1) The related work is not detailed. There are many graph neural networks to deal with low-pass properties of MPGCNs [1], [2].

(2) I think the novelty of this paper is limited. The adaptive graph filter is a not novel method. For example,  GPR-GNNs [3] proposes adaptive polynomial graph filtering, which is similar to the proposed method. Please comment the difference.

(3) The evaluation for the proposed methods is not sufficient. It is highly recommended to conduct node classification experiments to validate BankGCN. There are node classification datasets, which are dependent on high-frequency information not low-frequency information. *i.e.,* texas, wisconsin, and cornell. [2],[4]

*[1] Scattering gcn: Overcoming oversmoothness in graph convolutional networks, Min, Yimeng et al., NeurIPS 2020.*

*[2] Beyond low-frequency information in graph convolutional networks, Bo, Deyu, et al., AAAI 2021.*

*[3] Adaptive universal generalized pagerank graph neural network, Chien, Eli, et al., ICLR 2021.*

*[4] Geom-gcn: Geometric graph convolutional networks, Pei, Hongbin, et al., ICLR 2020.*

**Summary Of The Paper:**

This paper proposes BankGCN, which simplifies spectral graph neural networks by utilizing an adaptive filter bank to extend the capabilities of GCNs beyond low-pass features. Compared to existing spectral graph convolutional networks, which have numerous free parameters, the proposed method reduces parameters by sharing learnable filters. The empirical study validates that BankGCN shows good performance on the graph classification task.

**Summary Of The Review:**

Overall, I recommend marginally below the acceptance score. My concern is about the limited novelty and not enough experimental results. Hopefully, the authors can address it.

---

> ### Author Response · Authors · 2021-11-23
> **Responses to Reviewer nrop:**
>
> We would like to thank you for the valuable comments.
>
> 1. More discussion of related works.
>
>    [A] Scattering GCN resorts to geometric scattering transform (predefined graph wavelets) to complement the low-pass features of GCN with band-pass features, and Bo et al. (2021) employ a one-order high-pass filter as the complement. Contrary to these methods, we employ a learnable filter bank with various frequency responses to adaptively capture diverse spectral characteristics of signals rather than merely adding high-pass features or specific components via predefined graph wavelets. Furthermore, we have provided more comparisons with related works in dealing with low-pass limitations in Section 2 and App. F in the revised version.
>
>
> 2. Novelty and comparison with adaptive polynomial graph filtering works, like GPR- GNNs.
>
>     [A] The proposed method clearly differs from existing methods, like GPR-GNNs. The whole network of GPR-GNNs corresponds to a high-order polynomial graph filter, whereas BankGCN, as a single convolution operator, corresponds to a filter bank composed of multiple such polynomial filters with diverse frequency responses. Thereby, BankGCN is more powerful in representing diverse frequency characteristics of graph signals. More comparisons with adaptive polynomial graph filtering works are provided in Section 2 and App. F in the revised version.
>
>     Furthermore, we consider and propose a sharing scheme between filters, which makes the number of free parameters nearly independent of the number of filters, and together with subspace projection effectively simplifies existing spectral methods. The proposed diversity regularization further guarantees the filters in a filter bank with different frequency characteristics in order to capture diverse spectral characteristics of graph data. These contributions demonstrate the novelty of our work. Additional comparisons are presented in Section 5.1 in the revised version that further support the effectiveness of BankGCN.
>
>
> 3. Conduct node classification experiments on datasets that are dependent on high-frequency information not low-frequency information. i.e., texas, wisconsin, and cornell.
>
>     [A] We have further conducted experiments in node classification tasks as suggested on texas, wisconsin, cornell, and actor datasets, in Section 5.1. As compared in Table 1, the proposed BankGCN outperforms all the baseline models, including a collection of recent works focusing on the over-smoothing issue, like MixHop (Abu-El-Haija et al., 2019), GEOM-GCN (Pei et al., 2020), H2GCN (Zhu et al., 2020), SGF (NT et al., 2021), etc. More results are presented in Section 5.1 in the revised version.

---

### Official Review · Reviewer_1kMb · 2021-11-02

**Correctness:** 1
**Technical Novelty And Significance:** 1
**Empirical Novelty And Significance:** 2
**Recommendation:** 3
**Confidence:** 4

**Main Review:**

Strong points: The ablation studies are quite comprehensive and give a good sense of the effects of the K, s, and \lambda parameters.

The Figures are clear and informative, the notation and writing is clear.

Weak points:

BankGCN tackles a by now well characterized problem, that of oversmoothing in the original formulations of graph convolutional networks. I find the claim that BankGCN is more powerful than "most" MPGCNs to be somewhat misleading, while true for those mentioned in the introduction, there are many new MPGCN architectures that address this problem, (mentioned below) as well as Scattering GCN and GPR-GNN (as mentioned in section 2). Comparison to at least some of these methods, or justification as to why these methods are not comparable is a must in an empirically driven paper like this.

While the experiments are well designed, although code is not provided, the improvements are relatively minor as compared to the variation inherent in these datasets (as seen in similar work). I would encourage the authors to try different datasets where more substantial differences between methods may emerge, as on these benchmarks improvement is difficult to show.

The "sharing scheme" amounts to a linear layer before message passing, I believe this has been used in a number of earlier works (although I could be convinced otherwise).

Questions:

It is mentioned that section 5.2 compares models with the same number of free parameters per hidden layer, how is this done and why was this done on different datasets (Cifar / molhiv)?

Other points:

I find the way the authors use the term MPGCNs a bit confusing, as claims are made about BankGCN in contrast to MPGCNs where BankGCN. is a message passing GCN architecture. Could this be clarified further in the writing?

While the notation is well explained, some of it is a bit unusual, for example \mathbf{\alpha} is a vector of \alpha coeffs, but R is a vector of \mathbf{r} features. T_k are chebyshev polynomials while T_\Omega(G, Y) is the cross-entropy target.

I believe the reference to Johannes Klicpera et al. (2019) in section 2 paragraph 2 is incorrect, you might have meant to reference the work:

Johannes Klicpera, Aleksandar Bojchevski, and Stephan Gunnemann. Predict then propagate: Graph neural networks meet personalized pagerank. In International Conference on Learning Representations, 2019

Rather than

Johannes Klicpera, Stefan Weißenberger, and Stephan Gunnemann. Diffusion improves graph learn- ing. In Adv. Neural Inf. Process. Syst. 32, pp. 13354–13366, 2019.

In fact the work that you reference employs a learnable filter bank contrary to your claim in section 2.

Additional recent work in

Hao Zhu and Piotr Koniusz. Simple Spectral Graph Convolution. In ICLR 2021.

Also tackles the problem of oversmoothing and learns polynomial filters of the Laplacian using a slightly different formulation.

**Summary Of The Paper:**

The authors introduce BankGCN, a graph convolutional network that learns (chebyshev) polynomial filters over the graph. BankGCN includes an initial subspace projection, and a cosine similarity regularization to encourage diverse filters. BankGCN is compared to a number of other graph networks on standard whole graph classification tasks.

The authors claim that most architectures of MPGCNs are limited in that they (1) focus on low frequency information and (2) lack a proper sharing scheme between filters, and that BankGCN addresses these two issues.

**Summary Of The Review:**


While the experiments are well designed, given that there are very few comparisons to similar works that claim to solve the problems of oversmoothing, and limited theoretical contribution I lean towards rejection.

---

> ### Author Response · Authors · 2021-11-23
> **Responses to Reviewer 1kMb：**
>
> We sincerely thank the reviewer for the helpful comments.
>
>
> 1. Comparison or discussion with some works tackling the over-smoothing issue.
>
>     [A] We have further provided comparisons with a series of works that tackle the over-smoothing issue, including MixHop (Abu-El-Haija et al., 2019), SGF (NT et al., 2021), ASGAT (Li et al., 2021), and H2GCN (Zhu et al., 2020), in both Related Work (Section 2 and App. F) and Experiments (Section 5.1).  In the additional node classification tasks presented in Table 1, BankGCN outperforms these recent baseline models and achieves the best performance. Furthermore, we have considered a variant of BankGCN, termed BankGCN-Diff, which predefines filters with graph diffusion wavelets (like Scattering GCN). As shown in Table 2, BankGCN-Diff underperforms on most datasets. These results validate the effectiveness of the proposed method in solving the over-smoothing issue.
>
>      1.1 Additional recent work S^2GC (Zhu & Koniusz, 2021).
>
>      [A] The whole network of S^2GC can be interpreted as a high-order polynomial filter, whereas BankGCN, as a convolution operator, consists of multiple such polynomial filters in the filter bank. Thereby, BankGCN is more powerful in handling diverse frequency characteristics of graph data. We have discussed it in Related Work in the revised version (Section 2 and App. F).
>
>
> 2. Try different datasets.
>
>     [A] Except for benchmark datasets, we have evaluated the models on the most recent and complex datasets, including Ogbg and CIFAR-10, where the proposed model outperforms all the baselines with a large margin, as presented in Table 4.
>
>
> 3. How and why compare models with the same number of free parameters per hidden layer ?
>
>     [A] It is implemented by adjusting the number of feature maps per hidden layer of different models to have the same number of free parameters per hidden layer. This experiment is complementary to the case that all of the baseline models have the same number of feature maps per hidden layer (Table 2), in order to further evaluate the expressiveness of models under the same degree of freedom.
>
>
> 4. A bit confusing of the using of the term MPGCNs.
>
>     [A] We have used “existing MPGCNs”or “classical MPGCNs” to clarify it in the revised version.
>
>
> 5. Notation.
>
>    [A] We generally use capital letters for matrices and bold lowercase letters for vectors. $\mathbf{\alpha}_{[p]}$ is a vector, while R denotes a feature matrix composed of $\mathbf{r}$. We have changed the target function from $T_\Theta(\mathcal{G}, Y)$ to $\mathscr{T}_\Theta(\mathcal{G}, Y)$ in order to distinguish it from $T_k$. Thanks for pointing it out.
>
>
> 6. Reference correction.
>
>     [A] Thanks, we have corrected it.

---

> > ### Author Response · Authors · 2021-11-23
> > **Responses to Reviewer 1kMb (Cont.)：**
> >
> > 7. "The "sharing scheme" amounts to a linear layer before message passing, I believe this has been used in a number of earlier works (although I could be convinced otherwise)."
> >
> >     [A] The sharing scheme relies to signal decomposition instead of just a linear layer.  Graph signals are first decomposed into different subspaces, and then the filter is shared within a subspace but differs across subspaces. In this paper, we employ a group of linear projections as an example of the signal decomposition, but more advanced decomposition methods are still suitable. We have not seen any related work in the consideration of the sharing scheme between filters in spectral GCNs.

---

### Official Review · Reviewer_yK19 · 2021-11-03

**Correctness:** 2
**Technical Novelty And Significance:** 2
**Empirical Novelty And Significance:** 3
**Recommendation:** 5
**Confidence:** 3

**Main Review:**

The manuscript proposes an interesting point of view on graph convolutional operators. The proposed idea is clearly defined and theoretically grounded. The empirical evaluation of the operator shows interesting results, but some points about the experimental methodology have to be clarified since the comparison with the baseline models seems not completely fair.

In the introduction, the authors discuss the spectral convolution networks, and they claim that the lack of proper sharing schemes between filters makes these models redundant and even prone to overfitting. To me, this point is not completely clear, and in particular the criticism about the sharing scheme. Since it is one of the motivations on which BankGCN relies, in my opinion, this point it should be extended and further discussed.

In section 4.2 the authors state that f_{[p]} are defined as linear projections. Moreover, all the functions project the input in subspaces that have the same dimension. In the performed experiments how similar are the projections computed by the various f_{[p]} have the same dimension? Does their similarity influence the expressiveness of the model?

The definition of the filters (section 4.3) recalls the one defined by Wu, Felix, et al. in  "Simplifying graph convolutional networks." (SGC) and the LGC proposed in (Navarin, Erb, et al. “Linear Graph Convolutional Networks.”), that exploits a normalized Laplacian instead of the Chebyshev polynomial. Considering that these models are very similar to the BankGCN with p=1, they should be also considered in the discussion and the experimental comparison.

For what concerns the experimental comparison to me it is not clear how the various models were validated, and so it is difficult to assess whether the reported comparison is fair or not. Indeed in section 5, the authors state:
“The network in the experiment consists of four convolution layers, one graph-level readout module and a linear classier.”
while in appendix C they state: “The trained model with the best validation performance is selected for test.” and on the next page: “...the learning rate is 0.001 and the batch size is 64. The number of training epochs is set as 500, and early stopping is employed with patience 30. Finally, we obtain the following optimal “hyper-parameters through grid search: weight decay ∈{0,1e−5,1e−4}and \gamma ∈{0,0.1,10}.”
The fact that the model with the best performance on the validation set was selected is of course correct, but my understanding only the weights decay and \gamma were selected through a grid search process while the other hyper-parameters seem to be set without performing any validation procedure.  Moreover, it is not clear whether the validation procedure was also used to select the hyper-parameters of the baselines models. Considering models that are all validated following the same validation procedure is crucial to perform a fair comparison. Indeed, as shown in (Errica et al.  “A Fair Comparison of Graph Neural Networks for Graph Classification”') the policy used to validate the hyperparameters of a model highly impacts the model's performance. The authors also state that “For a fair comparison, the results of baseline models are obtained with the same configurations as BankGCN using the public versions provided in the pytorch-geometric package”. In my opinion, it is not a fair way to compare the models using the configuration selected for the BankGCN,  mainly because each model has to be validated independently, in order to select the hyper-parameters that ensure the model to achieve the best performance (on the validation set).
In section 5 “For a fair and complete comparison, we consider two cases, (i) the same number of features, i.e., all of the models with the same number of feature maps per hidden layer, and (ii) the same number of parameters, i.e., models with nearly the same number of free parameters per hidden layer”. Comparing the models that exploit the same number of parameters is a very interesting analysis and highlights the expressiveness of the models but it does not show how good each model is in the graph classification task (as said before, this evaluation would require validating each model independently).

The results reported in Table 1 show that, in many cases, the accuracy gains obtained by the proposed method seem within noise levels due to high variance. Therefore it would be better to perform a statistical significance test,  to identify which improvements are significant. Note that the methodology followed to obtain the results and validation policy influence also the variance of the accuracy detected for each method.

Minors:
eq (6): the last parenthesis is missing
the acronym IGFT is used without a prior definition of its meaning
For what concerns the paper organization, putting the notation in the appendix makes it a bit complex for the reader to follow the mathematical definition of the various components. in my opinion, the notation definition has to be present in the paper before the start of the theoretical discussion.


**Summary Of The Paper:**

The manuscript proposes a novel convolutional operator dubbed BankGCN, that given a node input signal, first projects it in different subspaces using a group of projection functions (one for each subspace), and then applies to each projection an adaptive filter.
As projection functions, the authors use Linear projections. The model was tested on the Graph classification task.

**Summary Of The Review:**

The manuscript proposes an interesting point of view on graph convolutional operators. The proposed idea is clearly defined and theoretically grounded. The empirical evaluation of the operator shows interesting results, but some points about the experimental methodology have to be clarified since the comparison with the baseline models seems not completely fair.

---

> ### Author Response · Authors · 2021-11-23
> **Responses to Reviewer yK19：**
>
> We would like to thank you for your valuable comments. We provide detailed responses to your questions below.
>
> 1. Clarification of the experimental methodology.
>
>    1.1  Hyper-parameters.
>
>     [A] We take the weights decay and $\gamma$ as hyper-parameters that are obtained through grid-search for each model independently. The other hyper-parameters, like network architecture, are fixed across the datasets for all the models without tuning for any model, similarly to previous works (Hu et al., 2020).
>
>    1.2  “It is not clear whether the validation procedure was also used to select the hyper-parameters of the baseline models.”
>
>     [A] Yes, the hyper-parameters (weight decay) are selected through grid-search using the validation procedure for different models, respectively. For the specific hyper-parameters of different models, we adopt the optimal settings provided in their original public papers, such as the number of heads in GAT and the MLP layers in GIN. We have clarified it in the revised version.
>
>    1.3 “Comparing the models that exploit the same number of parameters is a very interesting analysis and highlights the expressiveness of the models but it does not show how good each model is in the graph classification task (as said before, this evaluation would require validating each model independently).”
>
>    [A] Each model is validated independently for its respective hyper-parameters. Thereby, this experiment can further demonstrate the expressiveness of different models under the same number of free parameters per hidden layer.
>
>    1.4 Perform a statistical significance test.
>
>    [A] To alleviate the impact of randomness, we already run the experiments with 20 randomly data splits and network initializations with different random seeds, and we report the mean and standard variation of results; this is consistent with most previous methods (Ma et al., 2019, Velikovi et al., 2018). The better mean results can support the effectiveness of the proposed model. Furthermore, BankGCN outperforms all of the baselines with a much larger margin than the standard variations, as shown in Table 4.
>
>
> 2. Further discussion of sharing schemes between filters.
>
>    [A] As discussed in Section 3, existing spectral methods like ChebNets employ a different filter for each mapping from an input channel to an output channel and adopt a total of d $\times$ d’ K -order polynomial filters (d and d’ denote the respective number of channels of input and output signals). In contrast, BankGCN first decomposes input signal to s subspaces, within each of which a filter is used to capture the frequency characteristics of signal components (i.e., a subspace shares a filter), and thereby a total of s filters are used. In contrast with the number of free parameters of existing spectral methods, namely O(d $\times$ d’ $\times$ (K+1)), that of BankGCN is only O(d $\times$ d’ + s $\times$ (K+1)). More theoretical analysis has been provided in Section 4.5.
>
>
> 3. “All the functions project the input in subspaces that have the same dimension. How similar are the projections computed by the various $f_{[p]}$ have the same dimension? Does their similarity influence the expressiveness of the model?”
>
>    [A] The simple case where subspaces have the same dimension is adopted as an example in this paper. It could be easily extended to the case with different dimensions. The projection functions are learned from data, and thereby their similarities adapt to datasets. Even if some projections may be similar partly, their respective filters are guaranteed to be different with the proposed diversity regularization, and thereby the output representations of different subspaces are diverse.
>
>
> 4. Comparison with SGC (Wu et al., 2019) and LGC (Navarin et al., 2020).
>
>    [A] We have discussed with SGC and LGC in related works (Section 2 and App. F), and further provided comparisons with SGC in node classification tasks, as presented in Table 1. Specifically, the whole network of SGC and LGC can be interpreted as a K-order polynomial filter, whereas BankGCN consists of multiple such K-order polynomial filters in the filter bank per layer. In other words, the whole network of SGC and LGC can be interpreted as a special case of BankGCN (a single convolution layer) with only one filter in the filter bank (s=1). Thereby, BankGCN is more powerful and flexible than SGC and LGC in dealing with diverse frequency characteristics of graph signals, as validated by the experimental results in Table 1.
>
>
> 5. Minors and notations.
>
>    [A] We have provided the full name and definition of IGFT (Eq. S2 in App. A). Due to space limitation, we have to move the notations to appendixes but we insert a link before at the start of Section3.
>
>
> Reference:
>
> Ma Y, Wang S, Aggarwal C C, et al. Graph convolutional networks with eigenpooling[C]//Proceedings of the 25th ACM SIGKDD International Conference on Knowledge Discovery & Data Mining. 2019: 723-731.

---

### Decision · Program_Chairs · 2022-01-20

**Decision:**

Reject

**Comment:**

The paper proposes a graph convolution operator (BankGCN) to be used in graph neural networks. The reviewers mainly raised concerns about the limited of novelty in the light of numerous previous works that are similar or address similar problems as well as lacking evaluation. While the rebuttal addressed some of the concerns, the overall impression is that the paper is not of sufficient methodological or experimental significance for the conference.